# Explaining Off-Policy Actor-Critic From A Bias-Variance Perspective

## Abstract

Off-policy Actor-Critic algorithms have demonstrated phenomenal experimental performance but still require better explanations. To this end, we show its policy evaluation error on the distribution of transitions decomposes into: a Bellman error, a bias from policy mismatch, and a variance term from sampling. By comparing the magnitude of bias and variance, we explain the success of the Emphasizing Recent Experience sampling and 1/age weighted sampling. Both sampling strategies yield smaller bias and variance and are hence preferable to uniform sampling.

## 1 Introduction

A practical reinforcement learning (RL) algorithm is often in an actor-critic setting (Lin, 1992; Precup et al., 2000) where the policy (actor) generates actions and the Q/value function (critic) evaluates the policy's performance. Under this setting, off-policy RL uses transitions sampled from a replay buffer to perform Q function updates, yielding a new policy $\pi$. Then, a finite-length trajectory under $\pi$ is added to the buffer, and the process repeats. Notice that sampling from a replay buffer is an offline operation and that the growth of replay buffer is an online operation. This implies off-policy actor-critic RL lies between offline RL (Yu et al., 2020; Levine et al., 2020) and on-policy RL (Schulman et al., 2015; 2017). From a bias-variance perspective, offline RL experiences large policy mismatch bias but low sampling variance, while on-policy RL has a low bias but high variance. Hence, with a careful choice of the sampling from its replay buffer, off-policy actor-critic RL may achieve a better bias-variance trade-off. This is the direction we explore this work.

To reduce policy mismatch bias, off-policy RL employs importance sampling with the weight given by the probability ratio of the current to behavior policy (the policy that samples the trajectories) (Precup et al., 2000; Xie et al., 2019; Schmitt et al., 2020). However, because the behavior policy is usually not given in practice, one can either estimate the probability ratio from the data (Lazic et al., 2020; Yang et al., 2020; Sinha et al., 2020) or use other reasonable quantities, such as the Bellman error (Schaul et al., 2016), as the sampling weight. Even using a naive uniform sampling from the replay buffer, some off-policy actor-critic algorithms can achieve a nontrivial performance (Haarnoja et al., 2018; Fujimoto et al., 2018). These observations suggest we need to better understand the success of off-policy actor-critic algorithms, especially in practical situations where a fixed behavior policy is unavailable.

Our contributions are as follows. To understand the actor-critic setting without a fixed behavior policy, we construct a non-stationary policy that generates the averaged occupancy measure. We use this policy as a reference and show the policy evaluation error in an off-policy actor-critic setting decomposes into the Bellman error, the policy mismatch bias, and the variance from sampling. Since supervised learning during the Q function update only controls the Bellman error, we need careful sampling strategies to mitigate bias and variance. We show that the 1/age weighting or its variants like the Emphasizing Recent Experience (ERE) strategy (Wang & Ross, 2019) are preferable because both their biases and variances are smaller than that of uniform weighting.

To ensure the applicability of our explanation to practical off-policy actor-critic algorithms, we adopt weak but verifiable assumptions such as Lipschitz Q functions, bounded rewards, and a bounded action space. We avoid strong assumptions such as a fixed well-explored behavior policy, concentration coefficients, bounded probability ratios (e.g., ratios of current to behavior policy), and tabular or linear function approximation. Hence our analysis is more applicable to practical settings. In

addition, our analysis suggests that 1/age-based samplings (e.g., ERE, ERE_apx, 1/age) have advantages in bias and variance. Our experiments verify that SAC (Haarnoja et al., 2018) with 1/age-based samplings outperform the prior work. Thus, our results not only provide theoretical foundations for practical off-policy actor-critic RL algorithms but also achieve better performances.

## 2 PRELIMINARIES

### 2.1 REINFORCEMENT LEARNING

Consider an infinite-horizon Markov Decision Process (MDP) $\langle \mathcal{S}, \mathcal{A}, T, r, \gamma \rangle$, where $\mathcal{S}$, $\mathcal{A}$ are finite-dimensional continuous state and action spaces, $r(s, a)$ is a deterministic reward function, $\gamma \in (0, 1)$ is the discount factor, and $T(s'|s, a)$ is the state transition density; i.e., the density of the next state $s'$ given the current state and action $(s, a)$. Given an initial state distribution $\rho_0$, the objective of RL is to find a policy $\pi$ that maximizes the $\gamma$-discounted cumulative reward when the actions along the trajectory follow $\pi$:

$$\max_\pi J(\pi) = \max_\pi \mathbb{E}\left[\sum_{i=0}^{\infty} \gamma^i r(s_i, a_i) \Big| s_0 \sim \rho_0, \ a_i \sim \pi(\cdot|s_i), \ s_{i+1} \sim T(\cdot|s_i, a_i)\right]. \quad (1)$$

Let $\rho_i(s|\rho_0, \pi, T)$ be the state distribution under $\pi$ at trajectory step $i$. Define the normalized state occupancy measure by

$$\rho_{\rho_0}^\pi(s) \triangleq (1 - \gamma) \sum_{i=0}^{\infty} \gamma^i \rho_i(s|\rho_0, \pi, T). \quad (2)$$

Ideally, the maximization in (1) is achieved using a parameterized policy $\pi_\theta$ and policy gradient updates with (Sutton et al., 1999; Silver et al., 2014):

$$\nabla_\theta J(\pi_\theta) = (1 - \gamma)^{-1} \mathbb{E}_{(s,a) \sim \rho_{\rho_0}^{\pi_\theta}}[(\nabla_\theta \log \pi_\theta(a|s)) Q^{\pi_\theta}(s, a)]. \quad (3)$$

Here $\rho_{\rho_0}^{\pi_\theta}(s, a) = \rho_{\rho_0}^{\pi_\theta}(s) \pi_\theta(a|s)$ is given in (2), and $Q^{\pi_\theta}$ is the Q function under policy $\pi_\theta$:

$$Q^\pi(s, a) = \mathbb{E}\left[\sum_{i=0}^{\infty} \gamma^i r(s_i, a_i) \Big| s_0 = s, \ a_0 = a, \ a_i \sim \pi(\cdot|s_i), \ s_{i+1} \sim T(\cdot|s_i, a_i)\right]. \quad (4)$$

Off-policy RL estimates $Q^\pi$ by approximating the solution of the Bellman fixed-point equation:

$$(\mathcal{B}^\pi Q^\pi)(s, a) = r(s, a) + \gamma \mathbb{E}_{s' \sim T(\cdot|s,a), \ a' \sim \pi(a|s)} Q^\pi(s', a') = Q^\pi(s, a).$$

It is well-known that $Q^\pi$ is the unique fixed point of the Bellman operator $\mathcal{B}^\pi$. Hence if $(B^\pi \hat{Q})(s, a) \approx \hat{Q}(s, a)$, then $\hat{Q}$ may be a "good" estimate of $Q^\pi$. In the next subsection, we will see that an off-policy actor-critic algorithm encourages this to hold for the replay buffer.

### 2.2 OFF-POLICY ACTOR-CRITIC ALGORITHM

We study an off-policy actor-critic algorithm of the form shown in Alg. 1. In line 2, for episode index $e$, $\pi^e$ samples one trajectory of length $L$. The transitions in this trajectory are then added to the replay buffer. Since the policies for different episodes are distinct, the collection of transitions in the replay buffer are generally inconsistent with the current policy. Notice the trajectory is simply collected by $\pi^e$, not a perturbed version of $\pi^e$ (e.g., gaussian corrupted version of $pi^e$).

Line 3 is a supervised learning that minimizes the Bellman error of $\hat{Q}$ using gradient descents, making $(\mathcal{B}^{\pi^e} \hat{Q})(s, a) \approx \hat{Q}(s, a)$ for $(s, a)$ in the replay buffer. When this holds, we will prove that the "distance" between $\hat{Q}$ and $Q^{\pi^e}$ become smaller over the replay buffer. This is a crucial step for line 4 to be truly useful. In line 4, $Q^{\pi^e}$ is replaced by $\hat{Q}$, and the policy is updated accordingly.

In practice, line 3 is replaced by mini-batch updates (Fujimoto et al., 2018; Haarnoja et al., 2018) where the summation of $J(\hat{Q}_\phi)$ is approximated by a sum over the mini-batches. In section 5.2, we show that a uniform-weighted mini-batch sampling biases learning towards older samples; this motivates the need for a countermeasure.

---

**Algorithm 1** Off-policy Actor-critic Algorithm

---

**Require:** Parameterized policy $\pi^e = \pi_{\theta_e}$ and Q function estimate $\hat{Q}_\phi$. Learning rate $\alpha$.
1: **for** episode $e = 1, 2, ...$ **do**
2:  Sample a length-$L$ trajectory $\{(s_i^e, a_i^e)\}_{i=0}^{L-1}$ with initial distribution $\rho_0(s)$ and policy $\pi^e(a|s)$.

3:  Evaluate the Bellman error as $J(\hat{Q}_\phi) = \sum_{j=1}^{e} \sum_{i=0}^{L-1} \left( \hat{Q}_\phi(s_i^j, a_i^j) - (\mathcal{B}^{\pi^e} \hat{Q}_\phi)(s_i^j, a_i^j) \right)^2$ and
    update $\hat{Q}_\phi$ as $\phi \leftarrow \phi - \alpha \nabla_\phi J(\hat{Q}_\phi)$.
4:  Approximate Eq. (3) as $\nabla_\theta \hat{J}(\pi_\theta) = (1-\gamma)^{-1} \mathbb{E}_{(s,a) \sim \rho_{\rho_0}^{\pi_\theta}} [(\nabla_\theta \log \pi_\theta(a|s)) \hat{Q}(s,a)]$ and up-
    date as $\theta_{e+1} = \theta_e + \alpha \nabla_\theta \hat{J}(\pi_\theta)$.
5: **end for**

---

Note that Alg. 1 seems to be inconsistent in the horizon because the Q function, Eq. (4), is defined in the infinite horizon but the trajectories are finite-length. In Corollary 1, we will address this inconsistency by approximating the Q function using finite-length trajectories.

## 2.3   THE CONSTRUCTION OF OUR BEHAVIOR POLICY

Although Alg. 1 doesn't have a fixed behavior policy, in Lemma 1, we construct a non-stationary policy that generates the averaged occupancy measure at every step. This describes the *averaged-over-episode behavior* at every trajectory step $i$ of the historical trajectory $\{(s_i^e, a_i^e)\}_{i,e=0,1}^{L-1,N}$. We hence define it to be our behavior policy and use it as a reference when analyzing Alg. 1.

The construction is as follows. Denote the state distribution at trajectory step $i$ in episode $e$ as $\rho_i^e(s) = \rho_i(s|\rho_0, \pi^e, T)$. By Eq. (2), $\rho_{\rho_0}^{\pi^e}$ is the state occupancy measure generated by $(\rho_0, \pi^e, T)$. Intuitively, $\rho_{\rho_0}^{\pi^e}$ describes the discounted state distribution starting at trajectory step 0 in episode $e$.

More generally, define $\rho_{\rho_i^e}^{\pi^e}$ as the state occupancy measure generated by $(\rho_i^e, \pi^e, T)$. Then, $\rho_{\rho_i^e}^{\pi^e}$ describes the discounted state distribution starting at trajectory step $i$ in episode $e$.

$$\rho_{\rho_i^e}^{\pi^e}(s) \triangleq (1-\gamma) \sum_{j=i}^{\infty} \gamma^{j-i} \rho_j^e(s) \tag{5}$$

**Behavior policy through averaging.**   Since Alg. 1 considers trajectories from all episodes, the average-over-episodes distribution $\varrho^{(i)} = N^{-1} \sum_e \rho_{\rho_i^e}^{\pi^e}$ will be of interest. Namely, $\varrho^{(i)}$ is the average of all occupancy measures starting at step $i$. To describe the average-over-episodes behavior at step $i$, we want a policy that generates $\varrho^{(i)}$, and Eq. (5) helps construct such a policy. Concretely, let $\overline{\rho}_i, \pi_i^D(a|s)$ be the averaged state distribution and the averaged policy at step $i$, respectively.

$$\varrho^{(i)}(s) \triangleq N^{-1} \sum_{e=1}^{N} \rho_{\rho_i^e}^{\pi^e}(s), \quad \overline{\rho}_i(s) \triangleq N^{-1} \sum_{e=1}^{N} \rho_i^e(s), \quad \pi_i^D(a|s) \triangleq \frac{\sum_{e=1}^{N} \pi^e(a|s) \rho_{\rho_i^e}^{\pi^e}(s)}{\sum_{e=1}^{N} \rho_{\rho_i^e}^{\pi^e}(s)}. \tag{6}$$

Lemma 1 shows $\pi_i^D$ in Eq. (6) is a notion of behavior policy in the sense that $(\overline{\rho}_i, \pi_i^D, T)$ generates $\varrho^{(i)}$; i.e., $\pi_i^D$ generates the averaged occupancy measures when the initial state follows $\overline{\rho}_i$. Since $\pi_i^D$ describes the averaged discounted behavior starting at step $i$, we define it to be our behavior policy. This is a key to analyzing the policy evaluation error in Alg. 1.

**Lemma 1.** *Let $\rho_{\overline{\rho}_i}^{\pi_i^D}(s)$ be the normalized state occupancy measure generated by $(\overline{\rho}_i, \pi_i^D, T)$. Then $\varrho^{(i)}(s) = \rho_{\overline{\rho}_i}^{\pi_i^D}(s)$ a.e. and hence from Eq. (6), $N^{-1} \sum_{e=1}^{N} \rho_{\rho_i^e}^{\pi^e}(s) \pi^e(a|s) = \rho_{\overline{\rho}_i}^{\pi_i^D}(s) \pi_i^D(a|s)$ a.e.*

## 3   RELATED WORK

There are two main approaches to off-policy RL: importance sampling (IS) (Tokdar & Kass, 2010) and regression-based approach. IS has a low bias but high variance, while the opposite holds for the regression-based approach. Below, we briefly review these techniques.

**Importance Sampling.** Standard IS uses behavior policy to form an importance weight from the probability ratio of the current to the behavior policy. For a fully accessible behavior policy, examples of this approach include: the naive IS weight (Precup et al., 2000), importance weight clipping (Schmitt et al., 2020) and the marginalized importance weight (Xie et al., 2019; Yin & Wang, 2020; Yin et al., 2021). Alternatively, one can use the density ratio of the occupancy measures of the current to the behavior policies (Liu et al., 2018). This is estimated using a maximum entropy approach (Lazic et al., 2020), a Lagrangian approach (Yang et al., 2020), or a variational approach (Sinha et al., 2020). A distinct approach emphasizes experience without considering probability ratios. Examples include emphasizing samples with a higher TD error (Schaul et al., 2016; Horgan et al., 2018), emphasizing recent experience (Wang & Ross, 2019) or updating the policy towards the past and discarding distant experience (Novati & Koumoutsakos, 2019). It is also shown that IS on replay buffer is equivalent to weighting on loss functions (Fujimoto et al., 2020).

**Regression-based.** A regression-based approach can achieve strong experimental results using proper exploration and function approximation (Fujimoto et al., 2018; Haarnoja et al., 2018). It also admits strong theoretical results such as generalization error using a concentration coefficient (Le et al., 2019), policy evaluation error using bounded probability ratios (Agarwal et al., 2019)[Chap 5.1], minimax optimal bound under linear function approximation Duan et al. (2020), confidence bounds constructed by kernel Bellman loss Feng et al. (2020). However, these settings require a fixed behavior policy and bounded probability ratios, which rarely hold in practice. We hence construct a non-stationary behavior policy to avoid this issue.

**Combined.** Prior work also suggests that combining IS and the regression-based yields robust results (Dudík et al., 2011; Jiang & Li, 2016; Thomas & Brunskill, 2016; Kallus & Uehara, 2020). It is also common to consider some refined contractions to improve stability. Examples include n-step Q-learning (Hessel et al., 2018), Retrace (Munos et al., 2016), Peng's Q (Kozuno et al., 2021), and other correction techniques (Harutyunyan et al., 2016; Tang et al., 2020; Rowland et al., 2020).

## 4 POLICY EVALUATION ERROR OF OFF-POLICY ACTOR-CRITIC ALGORITHMS

Let $Q^*$, $Q^{\pi^N}$ be the Q function of the optimal policy and $\pi^N$, respectively. Let $\hat{Q}$ be the estimated Q function. The performance error $|\hat{Q} - Q^*|$ decomposes into $|\hat{Q} - Q^{\pi^N}| + |Q^{\pi^N} - Q^*|$. The first term $|\hat{Q} - Q^{\pi^N}|$ is the policy evaluation error and is the focus in the off-policy evaluation literature (Duan et al., 2020). The second term is the policy's optimality gap and to bound this term currently requires strong assumptions such as tabular or linear MDPs (Jin et al., 2018; 2020). In an off-policy actor-critic setting, the policy evaluation error has not been analyzed adequately since most analysis requires a fixed behavior policy. This is the focus of our analysis.

Suppose we are given the trajectories sampled in the past episodes (the replay buffer). We analyze the policy evaluation error over the expected distribution of transitions. We express this error in terms of the Bellman error of $\hat{Q}$, the bias term in 1-Wasserstein distance between the policies $(\pi^N, \pi_i^D)$, and the variance term in the number of trajectories $N$. Note $\pi_i^D$ is the behavior policy at trajectory step $i$ defined in Eq. (6). The use of 1-Wasserstein distance (Villani, 2008) makes the results applicable to both stochastic and deterministic policies. Since supervised learning only makes the Bellman error small, we need good sampling strategies to mitigate the bias and variance terms. We hence investigate sampling techniques from a bias-variance perspective in the next section.

### 4.1 PROBLEM SETUP

**Notation.** In episode $e$, a length-$L$ trajectory $\{(s_i^e, a_i^e)\}_{i=0}^{L-1}$ following policy $\pi^e$ is sampled (Alg. 1, line 2). Then, $\hat{Q}$ is fitted over the replay buffer (line 3). Because the error of $\hat{Q}$ at step $i$ depends on the states sampled at steps $i, i+1, \ldots$, the importance of these samples $(s, a)$ depends on the trajectory step $i$. Also, due to the discount factor, the importance of step $j > i$ is discounted by $\gamma^{j-i}$ relative to step $i$. Hence, we will use a Bellman error and a policy mismatch error that reflects the dependency on the trajectory step and the discount factor. For $f : \mathcal{S} \times \mathcal{A} \to \mathbb{R}$ and $g : \mathcal{S} \to \mathbb{R}$, define an averaging-discounted operator over the replay buffer in $N$ episodes:

$$\tilde{E}_i^L f(\cdot, \cdot) \triangleq \frac{1}{N} \sum_{e=1}^{N} \sum_{j=i}^{L-1} (1-\gamma)\gamma^{j-i} f(s_j^e, a_j^e) \quad \text{and} \quad \tilde{E}_i^L g(\cdot) \triangleq \frac{1}{N} \sum_{e=1}^{N} \sum_{j=i}^{L-1} (1-\gamma)\gamma^{j-i} g(s_j^e)$$

Using $\tilde{E}_i^L$, the Bellman error of $\hat{Q}$ and the distance between the policies $(\pi^N, \pi_i^D)$ on the replay buffer are written as

$$
\begin{aligned}
\epsilon_{\hat{Q}}^{i,L} &= \tilde{E}_i^L \Big| \hat{Q}(\cdot, \cdot) - (\mathcal{B}^{\pi^N} \hat{Q})(\cdot, \cdot) \Big|, \\
W_1^{i,L} &= \tilde{E}_i^L W_1(\pi^N || \pi_i^D)(\cdot).
\end{aligned}
\tag{7}
$$

$W_1(\pi^N || \pi_i^D)(s) = W_1(\pi^N(\cdot|s) || \pi_i^D(\cdot|s))$ is the 1-Wasserstein distance between two policies at state $s$, which can be viewed as a function of $(s, a)$. Both the Bellman error $\epsilon_{\hat{Q}}^{i,L}$ and the policy mismatch error $W_1^{i,L}$ depend on trajectory step $i$ and are both discounted by $\gamma$.

**Assumptions.** We now relate the Bellman error and the policy mismatch error defined in Eq. (7) to the policy evaluation error $|\hat{Q} - Q^{\pi^N}|$. First, to control the error of $\hat{Q}$ by policy mismatch error in $W_1$ distance, we assume that for every state, $\hat{Q}$ and $Q^{\pi^N}$ are $L_{\mathcal{A}}$-Lipschitz over actions. We provide reasoning for this assumption in the later discussion. Next, observe that both quantities in Eq. (7) are random with sources of randomness from initial states, policies at different episodes, and the state transitions. To control this randomness, we need assumptions on the Q functions and apply a concentration inequality. Because Alg. 1 only samples *one* trajectory in each episode, higher-order quantities (e.g., variance) is unavailable. This motivates us to use first-order quantities (e.g., a uniform bound on the Q functions) and apply Hoeffding's inequality. Hence, we assume that $\hat{Q}(s, a)$ and $Q^{\pi^N}(s, a)$ are bounded in the interval $[0, r^{\max}/(1 - \gamma)]$ and that the action space is bounded with the diameter $\mathrm{diam}_{\mathcal{A}}$. A justification of these assumptions is provided in Appendix A.2.

## 4.2 POLICY EVALUATION ERROR

Observe that the Q function, Eq. (4), is defined on infinite-length trajectories while the errors, Eq. (7), are evaluated on length-$L$ trajectories. Discounting makes it possible to approximate the Q functions using finite-length $L$. To approximate the Q function at trajectory step $i$ up to a constant error, we need samples at least to step $i + \Omega((1 - \gamma)^{-1})$. Hence, we first prove the main theorem with $i = 0$ and $L = \infty$. Then, we generalize the result to $i \geq 0$ and finite $L$ in Corollary 1.

**Theorem 1.** *Let $N$ be the number of episodes. For $f : \mathcal{S} \times \mathcal{A} \to \mathbb{R}$, define the operator $\tilde{E}_0^\infty f \triangleq \frac{1}{N} \sum_{e=1}^N \sum_{i=0}^\infty (1 - \gamma)\gamma^i f(s_i^e, a_i^e)$. Denote the policy mismatch error and the Bellman error as $W_1^{0,\infty} = \tilde{E}_0^\infty W_1(\pi^N || \pi_0^D)(\cdot)$ and $\epsilon_{\hat{Q}}^{0,\infty} = \tilde{E}_0^\infty |\hat{Q}(\cdot, \cdot) - (\mathcal{B}^{\pi^N} \hat{Q})(\cdot, \cdot)|$, respectively. Assume*

   *(1) For each $s \in \mathcal{S}$, $\hat{Q}(s, a)$ and $Q^{\pi^N}(s, a)$ are $L_{\mathcal{A}}$-Lipschitz in $a$.*

   *(2) $\hat{Q}(s, a)$ and $Q^{\pi^N}(s, a)$ are bounded and take values in $[0, r^{\max}/(1 - \gamma)]$.*

   *(3) The action space is bounded with diameter $\mathrm{diam}_{\mathcal{A}} < \infty$.*

*Then, with probability at least $1 - \delta$,*

$$
\begin{aligned}
& \mathbb{E}_{(s_0, a_0) \sim \rho_0(s) \pi_0^D(a|s)} \Big| \hat{Q}(s_0, a_0) - Q^{\pi^N}(s_0, a_0) \Big| \\
\leq & \frac{r^{\max}}{1 - \gamma} \sqrt{\frac{1}{2N} \log \frac{2}{\delta}} + \Big( \frac{r^{\max}}{(1 - \gamma)^2} + \frac{2L_{\mathcal{A}} \mathrm{diam}_{\mathcal{A}}}{1 - \gamma} \Big) \sqrt{\frac{2}{N} \log \frac{2}{\delta}} + \frac{1}{1 - \gamma} \Big( \epsilon_{\hat{Q}}^{0,\infty} + 2L_{\mathcal{A}} W_1^{0,\infty} \Big).
\end{aligned}
$$

Theorem 1 expresses the error of $\hat{Q}$ as the sum of Bellman error, bias, and variance terms. To be more specific, the first two terms are understood as the "variance from sampling" because these decrease in the number of episodes $N$. On the other hand, $W_1^{0,\infty}$ is the "policy mismatch bias" w.r.t. $\pi^N$. Because the behavior policy $\pi_0^D$ is a mixture of the historical policies $\{\pi^e\}_{e=1}^N$, Eq. (6), we expect it to increase in $N$ until $\pi^N$ begins to converge.

Theorem 1 only indicates the difference between $\hat{Q}$ and $Q^{\pi^N}$ at $i = 0$ and infinite-length trajectories. We can generalize it to $i \geq 0$ and finite-length trajectories as follows. Recall from Eq. (6) and Lemma 1, the average state distribution at the $i$-th step, $\bar{\rho}_i$, and the behavior policy at the $i$-th step,

$\pi_i^D$, generate the average state occupancy measure, i.e., $\rho_{\overline{\rho}_i}^{\pi_i^D} = N^{-1} \sum_{e=1}^{N} \rho_{\rho_i^e}^{\pi_i^e}$ $a.e.$ Therefore, by restricting attention to the states sampled at time steps $i, i+1, \ldots$ the "initial state distribution" and the behavior policy become $\overline{\rho}_i$ and $\pi_i^D$, which generalizes Theorem 1 from $i = 0$ to $i \geq 0$. In addition, due to $\gamma$-discounting, we may use length-$L$ trajectories to approximate infinite-length ones, provided that $L \geq i + \Omega((1-\gamma)^{-1})$. These observations lead to the following corollary.

**Corollary 1.** *Fix assumptions (1) (2) (3) of Theorem 1. Rewrite the policy mismatch error and the Bellman error as $W_1^{i,L} = \tilde{E}_i^L W_1(\pi^N || \pi^D)(\cdot)$ and $\epsilon_{\hat{Q}}^{i,L} = \tilde{E}_i^L |\hat{Q}(\cdot, \cdot) - (\mathcal{B}^{\pi^N} \hat{Q})(\cdot, \cdot)|$, respectively. Note $\tilde{E}_i^L$ is defined in Eq. (7). Then, with probability at least $1 - \delta$,*

$$\mathbb{E}_{(s_i, a_i) \sim \overline{\rho}_i(s) \pi_i^D(a|s)} \left| \hat{Q}(s_i, a_i) - Q^{\pi^N}(s_i, a_i) \right| \leq$$

$$\frac{r^{\max}}{1-\gamma} \sqrt{\frac{1}{2N} \log \frac{2}{\delta}} + \left( \frac{r^{\max}}{(1-\gamma)^2} + \frac{2L_{\mathcal{A}} diam_{\mathcal{A}}}{1-\gamma} \right) \left( \sqrt{\frac{2}{N} \log \frac{2}{\delta}} + \gamma^{L-i} \right) + \frac{1}{1-\gamma} (\epsilon_{\hat{Q}}^{i,L} + 2L_{\mathcal{A}} W_1^{i,L}).$$

*Moreover, if $i \leq L - \frac{\log \epsilon}{\log \gamma}$ with $0 < \epsilon < 1$ and the constants are normalized as $L_{\mathcal{A}} = c/(1-\gamma)$, $\epsilon_{\hat{Q}}^{i,L} = \xi_{\hat{Q}}^{i,L}/(1-\gamma)$, then, with probability at least $1 - \delta$,*

$$\mathbb{E}_{(s_i, a_i) \sim \overline{\rho}_i(s) \pi_i^D(a|s)} \left| \hat{Q}(s_i, a_i) - Q^{\pi^N}(s_i, a_i) \right|$$

$$\leq \tilde{O} \left( \frac{1}{(1-\gamma)^2} \left( (r^{\max} + c \cdot diam_{\mathcal{A}})(1/\sqrt{N} + \epsilon) + \xi_{\hat{Q}}^{i,L} + c \cdot W_1^{i,L} \right) \right),$$

*where $\tilde{O}(\cdot)$ is a variant of $O(\cdot)$ that ignores logarithmic factors.*

Note that if $i = 0$ and $L = \infty$, Corollary 1 is identical to Theorem 1. Because the Bellman error of $\hat{Q}$ and the bias of the policy are both evaluated using the averaging-discounted operator $\tilde{E}_i^L$, Corollary 1 implies the difference of $\hat{Q}$ and $Q^{\pi^N}$ at trajectory step $i$ mainly depends on states at trajectory steps $\geq i$. Since the Q function is a discounted sum of rewards from the current step to the future, the error at step $i$ should mainly depend on the steps $\geq i$.

**Normalization.** Although the first conclusion in Corollary 1 gives a bound on the error of $\hat{Q}$, the constants may implicitly depend on the expected horizon $(1-\gamma)^{-1}$. Hence its interpretation requires care. For instance, $L_{\mathcal{A}}$, the Lipschitz constant of the Q functions w.r.t. actions, is probably the most tricky constant. While it is used extensively in the prior work (Luo et al., 2019; Xiao et al., 2019; Ni et al., 2019; Yu et al., 2020), its magnitude is never properly addressed in the literature. Intuitively, if a policy $\pi$ is good enough, it should quickly correct some disturbance on actions. In this case, the rewards after the disturbance only differ in a few trajectory steps, so the Lipschitzness of $Q^{\pi}$ in actions is sublinear in $(1-\gamma)^{-1}$. On the other hand, if the policy $\pi$ fails to correct a disturbance $\delta$, due to error propagation, the error $\delta$ propagates to every future step, leading to a linear error dependency to the horizon $O(\delta H)$. Therefore, the Lipschitzness of $Q^{\pi}$ in actions can be as large as $O((1-\gamma)^{-1})$. In addition to $L_{\mathcal{A}}$, the Bellman error $\epsilon_{\hat{Q}}^{i,L}$ should scale linearly in $(1-\gamma)^{-1}$ because the Q function represents the discounted cumulative reward, Eq. (4), which scales linearly in $(1-\gamma)^{-1}$. These observations suggest that $\epsilon_{\hat{Q}}^{i,L}$ and $L_{\mathcal{A}}$ in Corollary 1 are either linear in the horizon or lie between sublinear and linear. To better capture the dependency on the horizon, we normalize the constants and get the second conclusion.

**Interpretation.** The second conclusion shows the approximation error from infinite to finite-length trajectories is bounded by a constant for $i \leq L - \frac{\log \epsilon}{\log \gamma} \overset{\text{Taylor}}{\approx} L - \Omega((1-\gamma)^{-1})$, and will become harder to control for the higher $i$ due to the lack of samples. Besides, the variance term dominates when $N$ is small, while the bias term dominates at large $N$. Therefore, one may imagine that the training of Alg. 1 has two phases. At phase 1, the variance term dominates and decreases in $N$, so the learning improves quickly as more trajectories are collected. At phase 2, the bias term dominates, so the policy evaluation error becomes harder to improve and $\pi^N$ tends to converge.

## 5 PRACTICAL SAMPLING STRATEGIES

As previously mentioned, the supervised learning during the Q function update fails to control the bias and variance. We need careful sampling techniques during the sampling from the replay buffer to mitigate the policy evaluation error. In particular, Wang & Ross (2019) proposes to emphasize recent experience (ERE) because the recent policies are closer to the latest policy. We show below that the ERE strategy is a refinement of 1/age weighting and that both methods help balance the expected selection number of each training transition $(s, a)$. Balanced selection numbers reduce both the policy mismatch bias and sampling variance. Hence, this suggests the potential usefulness of ERE and 1/age, which we verify through experiments in the last subsection.

### 5.1 EMPHASIZING RECENT EXPERIENCE

In Wang & Ross (2019), the authors use a length-$K$ trajectory ($K$ may differ across episodes) and perform $K$ updates. In the $k$-th ($1 \leq k \leq K$) update, a mini-batch is sampled uniformly from the most recent $c_k = \max(N_0 \eta^{k\frac{L_0}{K}}, c_{\min})$ samples, where $N_0$ is the current size of the replay buffer, $L_0$ is the maximum horizon of the environment, $\eta$ is the decay parameter, and $c_{\min}$ is the minimum coverage of the sampling. For MuJoCo (Todorov et al., 2012) environments, the paper suggests the values: $(L_0, \eta, c_{\min}) = (1000, 0.996, 5000)$. One can see that $\eta = 1$ does a uniform weighting over the replay buffer, and the emphasis on the recent data becomes larger as $\eta$ becomes smaller. To see how the ERE strategy affects the mini-batch sampling, we prove the following result.

**Proposition 1.** *ERE is approximately equivalent (Taylor Approx.) to the non-uniform weighting:*

$$w_t \propto \frac{1}{\max(t, c_{\min}, N_0 \eta^{L_0})} - \frac{1}{N_0} + \frac{\mathbb{1}(t \leq c_{\min})}{c_{\min}} \max\left( \ln \frac{c_{\min}}{N_0 \eta^{L_0}}, 0 \right), \tag{8}$$

*where $t$ is the age of a data point relative to the newest time step; i.e., $w_1$ is the newest sample.*

Note that Prop. 1 holds for $\eta \neq 1$ because it is derived from the geometric series formula: $(1 - \eta^n)/(1 - \eta)$, which is valid when $\eta \neq 1$. Despite this discontinuity, we may still claim that the ERE strategy performs a uniform weighting when $\eta$ is close to 1. This is because when $\eta \approx 1$, Eq. (8) suggests $w_t$ is proportional to $1/(N_0 \eta^{L_0}) - 1/N_0$ for all $1 \leq t \leq N_0$, which is a uniform weighting. The emphasis on the recent experience (indicated by $\mathbb{1}(t \leq c_{\min})$) is also evident from Eq. (8). Precisely, the second term increases logarithmically $\ln \frac{1}{\eta}$ when $\eta$ becomes smaller, so the smaller $\eta$ indeed gives more weight on the recent experience.

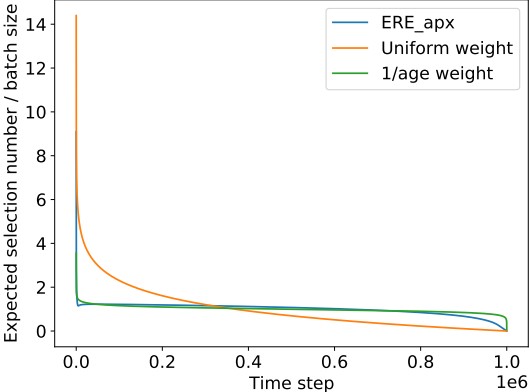

**Discussion.** A key distinction between the original ERE and Prop. 1 is that the original ERE considers the trajectory length $K$ while Prop. 1 doesn't. Intuitively, the disappearance of $K$'s dependency results from the aggregation of all effects in $1 \leq k \leq K$ updates. We verify that Eq. (8) tracks the original ERE well in the next subsection.

Another feature of Prop. 1 is an implicit 1/age weighting in Eq. (8). Although the original ERE *samples uniformly from the recent $c_k$ points*, the aggregate effect of all $1 \leq k \leq K$ updates appears to be well approximated by a 1/age weighting.

Figure 1: Expected selection numbers (aggregate weights) over a million time steps.

To understand the effect of 1/age, recall from section 2.2 that in practice, $\hat{Q}$ is updated using mini-batch samples. Define a point $(s_i^e, a_i^e)$'s time step as $i + 1 + L \cdot (e - 1)$. Then the expected number of times in all batch samples that a point at a certain time step is selected (the expected selection number) gives the aggregate weight over the time steps. As shown in Figure. 1, 1/age weighting and ERE_apx, Eq. (8), give almost uniform expected selection numbers across time steps while uniform weighting is significantly biased toward the old samples. Therefore, 1/age weighting and its variants help balance the expected selection number.

## 5.2 POLICY EVALUATION ERROR UNDER NON-UNIFORM WEIGHTS

Recall the expected selection numbers are the aggregate weights over time. To understand the merit of balanced selection numbers, we develop an error analysis under non-uniform aggregate weights in the Appendix A.5. In particular, Corollary 2 generalizes Corollary 1 to non-uniform weights $\{w_i > 0\}$ such that the policy evaluation error is bounded by three terms: Hoeffding error $\tilde{O}(\sqrt{\sum_i w_i^2}/\sum_i w_i)$ (weighted variance), weighted bias $W_{1,w}^{i,L}$, and normalized weighted bellman error $\xi_{\hat{Q},w}^{i,L}$. Since the weighted bellman error is well-controlled by supervised learning, in the following, we discuss the weighted biases and variances in ERE, 1/age weighting, and uniform sampling. We conclude that the ERE and 1/age weighting are better because both of their biases and variances are smaller than that of uniform sampling.

For the bias from policy mismatch, Figure 1 shows the uniform sampling (for each sampling from the replay buffer) makes the aggregate weights (expected selection number) bias toward old samples. Because the old policies tend to be distant from the current policy, the uniform sampling induces a larger weighted bias $W_{1,w}^{i,L}$ than ERE and 1/age do.

As for the variance from sampling, we generalize Corollary 1's uniform case: $\tilde{O}(1/\sqrt{N})$ to Corollary 2's weighted case: $\tilde{O}(\sqrt{\sum_i w_i^2}/\sum_i w_i)$. That is, the variance under non-uniform weight is bounded by the Hoeffding error $\tilde{O}(\sqrt{\sum_i w_i^2}/\sum_i w_i)$ and is reduced to $\tilde{O}(1/\sqrt{N})$ when the weights are equal. Furthermore, one can prove that the Hoeffding error is minimized under uniform weights:

**Proposition 2.** *Let $\{w_t > 0\}_{t=1}^N$ be the weights of the data indexed by $t$. Then the Hoeffding error $\sqrt{\sum_{t=1}^N w_t^2}/\sum_{t=1}^N w_t$ is minimized when the weights are equal: $w_t = c > 0, \ \forall \ t$.*

Therefore, the variance from sampling is large under non-uniform aggregate weights and is minimized by uniform aggregate weights. Since the uniform sampling leads to more non-uniform aggregate weights than ERE and 1/age weighting, its variance is also larger.

Because the ERE and 1/age weighting have balanced selection numbers (i.e., balanced aggregate weights), their biases and variance are smaller. They should perform better than uniform weighting for off-policy actor-critic RL. We will verify this in the next subsection.

## 5.3 EXPERIMENTAL VERIFICATION

Since we've established a theoretical explanation for 1/age-based samplings (ERE, ERE_apx, 1/age), we will explore two main propositions from the preceding subsections: (1) Are 1/age-based samplings better than uniform weighting? (2) Does the approximated ERE proposed in Eq. (8) track the original ERE well? In addition, since prioritized experience replay (Schaul et al., 2016) (PER) is a popular sampling method, a natural question is (3) Do 1/age-based samplings outperform PER?

We evaluate five sampling methods (ERE, ERE_apx, 1/age weighting, uniform, PER) on five MuJoCo continuous-control environments (Todorov et al., 2012): Humanoid, Ant, Walker2d, HalfCheetah, and Hopper. All tasks have a maximum horizon of 1000 and are trained using a Pytorch Soft-Actor-Critic (Haarnoja et al., 2018) implementation on Github (Tandon, 2018). Because the standard SAC implementation uses the uniform sampling, by comparing uniform sampling with the other four methods, we can deduce ways to boost SAC from a sampling's perspective.

Most hyper-parameters of the SAC algorithm are the same as that in Tandon (2018) except for the batch size, where we find a batch size of 512 tends to give more stable results. Our code is available at `https://github.com/sunfex/weighted-sac`. The SAC implementation and the MuJoCo environment are licensed under the MIT license and the personal student license, respectively. The experiment is run on a server with an Intel i7-6850K CPU and Nvidia GTX 1080 Ti GPUs.

In Figure 2, 1/age-based samplings (ERE, ERE_apx, 1/age) perform better than the uniform weighting does in all tasks. This verifies our preceding assertion that 1/age-based samplings are superior because their biases and variances of the estimated Q function are smaller. Moreover, ERE and ERE_apx mostly coincide with each other, so Eq. (8) is indeed a good approximation of the ERE strategy. This also explains the implicit connection between ERE and 1/age weighting strategies: ERE is almost equivalent to ERE_apx and 1/age is the main factor in ERE_apx, so ERE and 1/age

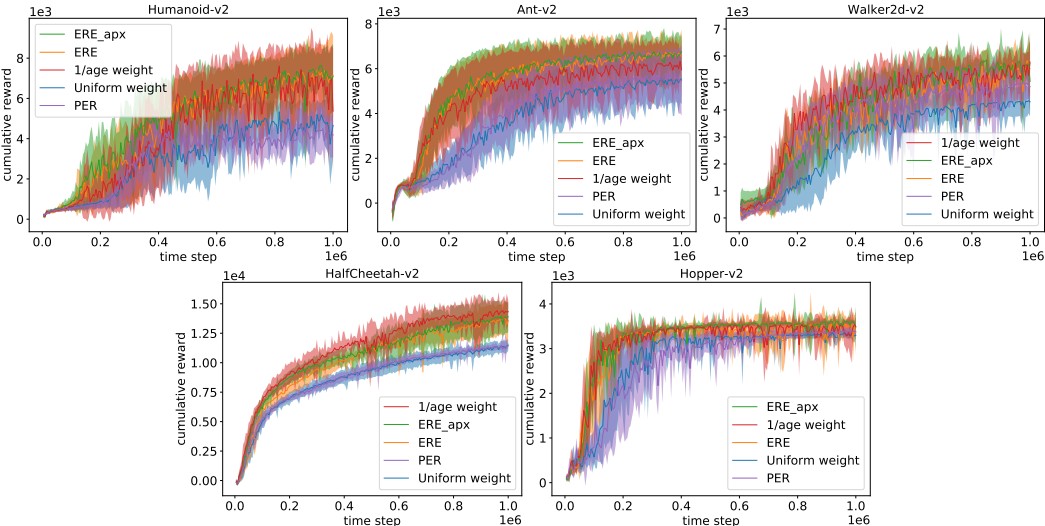

Figure 2: ERE, ERE_apx, 1/age weighting, PER, and uniform weighting on MuJoCo environments in a million steps. The solid lines and shaded areas are the means and one standard deviations.

weighting should produce similar results. Finally, ERE and 1/age generally outperform PER, so the 1/age-based samplings that we study achieve nontrivial performance improvements.

Finally, we provide performance comparison with the prior work Fujimoto et al. (2020); Kozuno et al. (2021) in Table 1). Table 1 shows the SAC with 1/age-based samplings (ERE, ERE_apx, 1/age) outperform the prior work. This suggests 1/age-based samplings not only possess simplicity and theoretical explanations but also achieve better performances.

| Methods | Hopper | HalfCheetah | Ant | Walker2D | Humanoid |
|---|---|---|---|---|---|
| SAC+ERE | 3402 | 13591 | 6516 | 5530 | **7147** |
| SAC+ERE_apx | 3477 | 13890 | **6757** | 5460 | 7088 |
| SAC+1/age | **3496** | **14328** | 5950 | **5774** | 6664 |
| TD3+LAP (Fujimoto et al., 2020) | 3364 | 10769 | 5593 | 4203 | 5445 |
| TD3+PAL (Fujimoto et al., 2020) | 3055 | 10584 | 4662 | 4458 | 5328 |
| TD3+Peng (Kozuno et al., 2021) | X | X | 4196 | 3675 | X |

Table 1: Performance comparison on MuJoCo environments at step 1 million. The numbers are measured from Fujimoto et al. (2020)[Figure 1] and Kozuno et al. (2021)[Figure 4]. The X symbol means the data are unavailable from the paper.

## 6   CONCLUSION

To understand off-policy actor-critic algorithms, we show the policy evaluation error on the expected distribution of transitions decomposes into the Bellman error, the bias from policy mismatch, and the variance from sampling. We use this to explain that a successful off-policy actor-critic algorithm should have a careful sampling strategy that controls its bias and variance. Motivated by the empirical success of Emphasizing Recent Experience (ERE), we prove that ERE is a variant of the 1/age weighting. We then explain that 1/age-based samplings (e.g., ERE, ERE_apx, 1/age) have smaller bias and variance and are preferable over uniform sampling. Our experiments verify that soft actor-critic with 1/age-based samplings outperforms the prior work. We hence conclude that a simple but careful design in the sampling of off-policy actor-critic RL can lead to better performances.

# 7 ETHICS STATEMENT

Our work explains off-policy actor-critic RL algorithms and designs simple but effective training techniques. The results apply to real domains where past trajectories are abundant while new samples are costly, e.g., robotics, recommender systems, and power grid management. RL algorithms may have positive economic effects by boosting efficiency and lowering risk. But inappropriate use can have negative societal impacts. These include job loss due to automation, the ethical challenges of delegating important decisions to a machine, and data privacy issues (e.g., recommendation agents). These implications apply to most control and RL studies and are not associated with any specific work.

# 8 REPRODUCIBILITY STATEMENT

To ensure reproducibility of the experiments, we've provided hyper-parameters in Appendix A.1 and the source code at an anonymous Github page `https://github.com/sunfex/weighted-sac`. Discussions on the assumptions are in section 4.1 and justifications are in Appendix A.2. A complete proof of the theorems and lemmas can be found in Appendix.

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

# A    Appendix

## A.1    Hyper-parameters

To train the SAC agents, we use deep neural networks to parameterize the policy and Q functions. Both networks consist of dense layers with the same widths. Table 2 presents the suggested hyper-parameters. As mentioned in the experiment section, the hyper-parameters are similar as the implementation in Tandon (2018).

## A.2    Justification of assumptions

In section 4.1, we introduce three main assumptions in this work. Below is a validation for each.

| Variable | Value |
|---|---|
| Optimizer | Adam |
| Learning rate | 3E-4 |
| Discount factor | 0.99 |
| Batch size | 512 |
| Model width | 256 |
| Model depth | 3 |

Table 2: Suggested hyper-parameters for SAC.

| Environment | Hopper | HalfCheetah | Walker2D | Ant | Humanoid |
|---|---|---|---|---|---|
| **Temperature** | 0.2 | 0.2 | 0.2 | 0.2 | 0.05 |

Table 3: Temperature parameters for SAC in MuJoCo environments.

1. **For each $s \in \mathcal{S}$, $\hat{Q}(s, a)$ and $Q^{\pi^N}(s, a)$ are $L_{\mathcal{A}}$-Lipschitz in $a$.** As mentioned in the paragraph "Normalization" of section 4.2, the Lipschitzness of $Q^{\pi^N}$ is sublinear or linear in the horizon, which quantifies the magnitude of $Q^{\pi^N}$'s Lipschitz constant. Since $\hat{Q}$ approximates $Q^{\pi^N}$, $\hat{Q}$ should have a similar property as long as the training error is well controlled. The practitioner can also enforce the Lipschitzness of $\hat{Q}$ using gradient penalty (Gulrajani et al., 2017) or spectral normalization (Miyato et al., 2018).

2. **$\hat{Q}(s, a)$ and $Q^{\pi^N}(s, a)$ are bounded and take values in $[0, r^{\max}/(1 - \gamma)]$.** This is a standard assumption in the RL literature. If the bound is violated, one can either clip, translate, or rescale to obtain new Q functions that satisfy the constraint. Note a bounded reward $r(s, a) \in [0, r^{\max}]$ has implied $Q^{\pi^N} \in [0, r^{\max}/(1 - \gamma)]$.

3. **The action space is bounded with diameter $\text{diam}_{\mathcal{A}} < \infty$.** This is a standard assumption in continuous-control environments and is usually satisfied in practice. It is also common to use clipping to ensure the bounds of the actions generated by the policy (Fujita & Maeda, 2018).

### A.3 THE CONSTRUCTION OF OUR BEHAVIOR POLICY

We first discuss some important relations between state occupancy measures and Bellman flow operator. Similar results about Fact 1 and 2 be found in Liu et al. (2018)[Lemma 3].

**Definition 1** (Bellman flow operator)**.** *The Bellman flow operator $B_{\rho_0, \pi, T}(\cdot)$ generated by $(\rho_0, \pi, T)$ with discount factor $\gamma$ is defined as*

$$B_{\rho_0, \pi, T}(\rho)(s) \triangleq (1 - \gamma)\rho_0(s) + \gamma \int T(s|s', a')\pi(a'|s')\rho(s')ds'da'.$$

**Fact 1.** *$B_{\rho_0, \pi, T}$ is a $\gamma$-contraction w.r.t. total variational distance.*

*Proof.* Let $p_1(s), p_2(s)$ be the density functions of some state distributions.

$$\begin{aligned}
D_{TV}(B_{\rho_0, \pi, T}(p_1) || B_{\rho_0, \pi, T}(p_2)) &= \frac{1}{2} \int \left| B_{\rho_0, \pi, T}(p_1(s)) - B_{\rho_0, \pi, T}(p_2(s)) \right| ds \\
&= \frac{1}{2} \int \gamma \left| \int T(s|s', a')\pi(a'|s')(p_1(s') - p_2(s'))ds'da' \right| ds \\
&\leq \frac{\gamma}{2} \int T(s|s', a')\pi(a'|s') \left| p_1(s') - p_2(s') \right| ds'da'ds \\
&= \frac{\gamma}{2} \int \left| p_1(s') - p_2(s') \right| ds' = \gamma D_{TV}(p_1 || p_2).
\end{aligned}$$

$\square$

**Fact 2.** *The normalized state occupancy measure $\rho_{\rho_0}^\pi$ generated by $(\rho_0, \pi, T)$ with discount factor $\gamma$ is a fixed point of $B_{\rho_0, \pi, T}$; i.e., $B_{\rho_0, \pi, T}(\rho_{\rho_0}^\pi)(s) = \rho_{\rho_0}^\pi(s)$.*

*Proof.*

$$\rho_{\rho_0}^{\pi}(s) = (1 - \gamma) \sum_{i=0}^{\infty} \gamma^i f_i(s|\rho_0, \pi, T)$$

$$= (1 - \gamma) f_0(s|\rho_0, \pi, T) + \gamma(1 - \gamma) \sum_{i=0}^{\infty} \gamma^i f_{i+1}(s|\rho_0, \pi, T)$$

$$= (1 - \gamma)\rho_0(s) + \gamma(1 - \gamma) \sum_{i=0}^{\infty} \gamma^i \int T(s|s', a')\pi(a'|s') f_i(s'|\rho_0, \pi, T) ds' da'$$

$$= (1 - \gamma)\rho_0(s) + \gamma \int T(s|s', a')\pi(a'|s')(1 - \gamma) \sum_{i=0}^{\infty} \gamma^i f_i(s'|\rho_0, \pi, T) ds' da'$$

$$= (1 - \gamma)\rho_0(s) + \gamma \int T(s|s', a')\pi(a'|s')\rho_{\rho_0}^{\pi}(s') ds' da' = B_{\rho_0, \pi, T}(\rho_{\rho_0}^{\pi})(s).$$

$\square$

Thus, the Bellman flow operator is useful to analyze the state occupancy measures, and we have the following lemma to construct the behavior policy $\pi_i^D$.

**Lemma 1.** *Let $\rho_i^e(s)$ the state distribution at trajectory step $i$ in episode $e$. Let $\rho_{\rho_i^e}^{\pi^e}(s)$ be the normalized occupancy measure starting at trajectory step $i$ in episode $e$. Then $\frac{1}{N} \sum_{e=1}^{N} \rho_{\rho_i^e}^{\pi^e}(s) = \rho_{\overline{\rho}_i}^{\pi_i^D}(s)$ a.e., where $\rho_{\overline{\rho}_i}^{\pi_i^D}$ is the normalized state occupancy measure is generated by $(\overline{\rho}_i, \pi_i^D, T)$. Moreover, we have $\frac{1}{N} \sum_{e=1}^{N} \rho_{\rho_i^e}^{\pi^e}(s)\pi^e(a|s) = \rho_{\overline{\rho}_i}^{\pi_i^D}(s)\pi_i^D(a|s)$ a.e.*

*Proof.* Precisely, $\rho_i^e(s) = \rho_i(s|\rho_0, \pi^e, T)$ is the state distribution at trajectory step $i$ following the laws of $(\rho_0, \pi^e, T)$. Since $\rho_{\rho_i^e}^{\pi^e}$ is the normalized occupancy measure generated by $(\rho_i^e, \pi^e, T)$, each $\rho_{\rho_i^e}^{\pi^e}$ is the fixed-point of the Bellman flow equation:

$$\rho_{\rho_i^e}^{\pi^e}(s) = (1 - \gamma)\rho_i^e(s) + \gamma \int T(s|s', a')\pi^e(a'|s')\rho_{\rho_i^e}^{\pi^e}(s') ds' da', \quad \forall e \in [1, ..., N].$$

This implies the average normalized occupancy measure is the fixed point of the Bellman flow equation characterized by $(\overline{\rho}_i, \pi_i^D, T)$:

$$\frac{1}{N} \sum_{e=1}^{N} \rho_{\rho_i^e}^{\pi^e}(s) = \frac{1}{N} \sum_{e=1}^{N} \left[ (1 - \gamma)\rho_i^e(s) + \gamma \int T(s|s', a')\pi^e(a'|s')\rho_{\rho_i^e}^{\pi^e}(s') ds' da' \right]$$

$$= (1 - \gamma)\overline{\rho}_i(s) + \gamma \int T(s|s', a')\frac{1}{N} \sum_{e=1}^{N} \left[ \pi^e(a'|s')\rho_{\rho_i^e}^{\pi^e}(s') \right] ds' da'$$

$$= (1 - \gamma)\overline{\rho}_i(s) + \gamma \int T(s|s', a')\frac{\sum_{e=1}^{N} \pi^e(a'|s')\rho_{\rho_i^e}^{\pi^e}(s')}{\sum_{e=1}^{N} \rho_{\rho_i^e}^{\pi^e}(s')}\frac{\sum_{e=1}^{N} \rho_{\rho_i^e}^{\pi^e}(s')}{N} ds' da'$$

$$= (1 - \gamma)\overline{\rho}_i(s) + \gamma \int T(s|s', a')\pi_i^D(a'|s')\frac{1}{N} \sum_{e=1}^{N} \rho_{\rho_i^e}^{\pi^e}(s') ds' da',$$

where $\pi_i^D(a|s) \triangleq \frac{\sum_{e=1}^{N} \pi^e(a|s)\rho_{\rho_i^e}^{\pi^e}(s)}{\sum_{e=1}^{N} \rho_{\rho_i^e}^{\pi^e}(s)}$ is the average behavior policy at step $i$. Since the Bellman flow operator is a $\gamma$-contraction in TV distance and hence has a unique fixed point in TV distance, denoted as $\rho_{\overline{\rho}_i}^{\pi_i^D}(s)$, we arrive at $\frac{1}{N} \sum_{e=1}^{N} \rho_{\rho_i^e}^{\pi^e}(s) = \rho_{\overline{\rho}_i}^{\pi_i^D}(s)$ almost everywhere. Also, by construction, we have

$$\frac{1}{N} \sum_{e=1}^{N} \rho_{\rho_i^e}^{\pi^e}(s)\pi^e(a|s) = \rho_{\overline{\rho}_i}^{\pi_i^D}(s)\pi_i^D(a|s) \text{ a.e.}$$

$\square$

### A.4 POLICY EVALUATION ERROR

**Lemma 2.** *If $Q(s, a)$ is $L_\mathcal{A}$-Lipschitz in $a$ for any $s$, then, for any state distribution $\rho$,*

$$\mathbb{E}_{s\sim\rho}\Big|\mathbb{E}_{a\sim\pi_1(\cdot|s)}Q(s,a) - \mathbb{E}_{a\sim\pi_2(\cdot|s)}Q(s,a)\Big| \leq L_\mathcal{A}\mathbb{E}_{s\sim\rho}W_1(\pi_1(\cdot|s)||\pi_2(\cdot|s)).$$

*Proof.* For any fixed $s$, we have

$$\mathbb{E}_{a\sim\pi_1(\cdot|s)}Q(s,a) - \mathbb{E}_{a\sim\pi_2(\cdot|s)}Q(s,a) = L_\mathcal{A}\Big[\mathbb{E}_{a\sim\pi_1(\cdot|s)}\frac{Q(s,a)}{L_\mathcal{A}} - \mathbb{E}_{a\sim\pi_2(\cdot|s)}\frac{Q(s,a)}{L_\mathcal{A}}\Big]$$

$$\leq L_\mathcal{A}\Big[\sup_{\|f\|_{\mathrm{Lip}}\leq 1}\mathbb{E}_{a\sim\pi_1(\cdot|s)}f(a) - \mathbb{E}_{a\sim\pi_2(\cdot|s)}f(a)\Big] = L_\mathcal{A}W_1(\pi_1(\cdot|s)||\pi_2(\cdot|s)),$$

where the second line follows from Kantorovich-Rubinstein duality (Villani, 2008). Since the 1-Wasserstein distance $W_1$ is symmetric, we can interchange the roles of $\pi_1$ and $\pi_2$, yielding

$$\Big|\mathbb{E}_{a\sim\pi_1(\cdot|s)}Q(s,a) - \mathbb{E}_{a\sim\pi_2(\cdot|s)}Q(s,a)\Big| \leq L_\mathcal{A}W_1(\pi_1(\cdot|s)||\pi_2(\cdot|s)).$$

Taking the expectation $\mathbb{E}_{s\sim\rho}$ on both sides completes the proof. $\square$

**Lemma 3.** *If $Q^\pi(s, a)$ is $L_\mathcal{A}$-Lipschitz in $a$ for any $s$, then*

$$\mathbb{E}_{(s_0,a_0)\sim\hat{\rho}_0(s)\pi_0^D(a|s)}\Big|Q^{\pi_0^D}(s_0,a_0) - Q^\pi(s_0,a_0)\Big| \leq \frac{L_\mathcal{A}}{1-\gamma}\mathbb{E}_{s\sim\rho_{\hat{\rho}_0}^{\pi_0^D}}W_1(\pi_0^D(\cdot|s)||\pi(\cdot|s)).$$

*Proof.* For any $(s, a)$, we have

$$|Q^{\pi_0^D}(s,a) - Q^\pi(s,a)|$$

$$= \gamma\Big|\mathbb{E}_{s'\sim T(\cdot|s,a)}\mathbb{E}_{\pi_0^D}Q^{\pi_0^D}(s',\pi_0^D(s')) - \mathbb{E}_\pi Q^\pi(s',\pi(s'))\Big|$$

$$\leq \gamma\mathbb{E}_{s'\sim T(\cdot|s,a)}\Big|\mathbb{E}_{\pi_0^D,\pi}Q^{\pi_0^D}(s',\pi_0^D(s')) - Q^\pi(s',\pi_0^D(s')) + Q^\pi(s',\pi_0^D(s')) - Q^\pi(s',\pi(s'))\Big|$$

$$\leq \gamma\mathbb{E}_{s'\sim T(\cdot|s,a)}\Big(\Big|\mathbb{E}_{\pi_0^D}Q^{\pi_0^D}(s',\pi_0^D(s')) - Q^\pi(s',\pi_0^D(s'))\Big| + \Big|\mathbb{E}_{\pi_0^D,\pi}Q^\pi(s',\pi_0^D(s')) - Q^\pi(s',\pi(s'))\Big|\Big)$$

$$\leq \gamma\mathbb{E}_{s'\sim T(\cdot|s,a)}\Big(\mathbb{E}_{\pi_0^D}\Big|Q^{\pi_0^D}(s',\pi_0^D(s')) - Q^\pi(s',\pi_0^D(s'))\Big| + L_\mathcal{A}W_1(\pi_0^D(\cdot|s')||\pi(\cdot|s'))\Big),$$

where the last line follows from Lemma 2. Let $\rho_i^{\pi_0^D}$ be the state distribution at step $i$ following the laws of $(\hat{\rho}_0, \pi_0^D, T)$. Take expectation over $\hat{\rho}_0$ and expand the recursive relation. We arrive at

$$\mathbb{E}_{(s_0,a_0)\sim\hat{\rho}_0(s)\pi_0^D(a|s)}\Big|Q^{\pi_0^D}(s_0,a_0) - Q^\pi(s_0,a_0)\Big| \leq L_\mathcal{A}\sum_{i=1}^\infty \gamma^i\mathbb{E}_{s_i\sim\rho_i^{\pi_0^D}}W_1(\pi_0^D(\cdot|s_i)||\pi(\cdot|s_i))$$

$$= \frac{L_\mathcal{A}}{1-\gamma}\mathbb{E}_{s\sim\rho_{\hat{\rho}_0}^{\pi_0^D}}W_1(\pi_0^D(\cdot|s)||\pi(\cdot|s)) - L_\mathcal{A}\mathbb{E}_{s\sim\hat{\rho}}W_1(\pi_0^D(\cdot|s)||\pi(\cdot|s))$$

$$\leq \frac{L_\mathcal{A}}{1-\gamma}\mathbb{E}_{s\sim\rho_{\hat{\rho}_0}^{\pi_0^D}}W_1(\pi_0^D(\cdot|s)||\pi(\cdot|s)),$$

where the second line follows from $\rho_{\hat{\rho}_0}^{\pi_0^D} = (1-\gamma)\sum_{i=0}^\infty \gamma^i\rho_i^{\pi_0^D}$. $\square$

**Lemma 4.** *If $Q(s, a)$ is $L_\mathcal{A}$-Lipschitz in $a$ for any $s$, then*

$$\mathbb{E}_{(s_0,a_0)\sim\hat{\rho}_0(s)\pi_0^D(a|s)}\Big|Q(s_0,a_0) - Q^{\pi_0^D}(s_0,a_0)\Big|$$

$$\leq \frac{\mathbb{E}_{(s,a)\sim\rho_{\hat{\rho}_0}^{\pi_0^D}(s)\pi_0^D(a|s)}\Big|Q(s,a) - \mathcal{B}^\pi Q(s,a)\Big| + L_\mathcal{A}W_1(\pi(\cdot|s)||\pi_0^D(\cdot|s))}{1-\gamma}.$$

*Proof.* For any $(s, a) \in \mathcal{S} \times \mathcal{A}$, we have a recursive relation:

$$\left|Q(s,a) - Q^{\pi_0^D}(s,a)\right| = \left|Q(s,a) - \mathcal{B}^{\pi_0^D} Q^{\pi_0^D}(s,a)\right|$$

$$\leq \left|Q(s,a) - \mathcal{B}^{\pi_0^D} Q(s,a)\right| + \left|\mathcal{B}^{\pi_0^D} Q(s,a) - \mathcal{B}^{\pi_0^D} Q^{\pi_0^D}(s,a)\right|$$

$$= \left|Q(s,a) - \mathcal{B}^{\pi_0^D} Q(s,a)\right| + \gamma \left|\mathbb{E}_{s' \sim T(\cdot|s,a), a' \sim \pi_0^D(\cdot|s')} Q(s',a') - Q^{\pi_0^D}(s',a')\right|$$

$$\leq \left|Q(s,a) - \mathcal{B}^{\pi_0^D} Q(s,a)\right| + \gamma \mathbb{E}_{s' \sim T(\cdot|s,a), a' \sim \pi_0^D(\cdot|s')} \left|Q(s',a') - Q^{\pi_0^D}(s',a')\right|$$

Expand the recursive relation. We have

$$\mathbb{E}_{(s_0,a_0) \sim \hat{\rho}_0(s)\pi_0^D(a|s)} \left|Q(s_0,a_0) - Q^{\pi_0^D}(s_0,a_0)\right|$$

$$\leq \mathbb{E}_{s_0 \sim \hat{\rho}_0} \mathbb{E}_{a_0 \sim \pi_0^D(\cdot|s_0)} \left|Q(s_0,a_0) - \mathcal{B}^{\pi_0^D} Q(s_0,a_0)\right|$$

$$+ \mathbb{E}_{s_0 \sim \hat{\rho}_0} \sum_{i=1}^{\infty} \gamma^i \mathbb{E}_{a_0, s_1, a_1, \ldots, s_{i-1}, a_{i-1}} \mathbb{E}_{s_i \sim T(\cdot|s_{i-1}, a_{i-1}), a_i \sim \pi_0^D(\cdot|s_i)} \left|Q(s_i, a_i) - \mathcal{B}^{\pi_0^D} Q(s_i, a_i)\right|$$

$$= \mathbb{E}_{s_0 \sim \hat{\rho}_0} \sum_{i=0}^{\infty} \gamma^i \mathbb{E}_{a_0, s_1, a_1, \ldots, s_i, a_i \sim T, \pi_0^D} \left|Q(s_i, a_i) - \mathcal{B}^{\pi_0^D} Q(s_i, a_i)\right|$$

$$= \frac{1}{1-\gamma} \mathbb{E}_{(s,a) \sim \rho_{\hat{\rho}_0}^{\pi_0^D}(s)\pi_0^D(a|s)} \left|Q(s,a) - \mathcal{B}^{\pi_0^D} Q(s,a)\right|. \tag{9}$$

The last line uses a fact for the normalized occupancy measure :

$$\mathbb{E}_{\rho_{\hat{\rho}_0}^{\pi_0^D}} = (1-\gamma) \left[ \mathbb{E}_{s_0 \sim \hat{\rho}_0} \mathbb{E}_{a_0 \sim \pi_0^D(\cdot|s_0)} + \sum_{i=1}^{\infty} \gamma^i \mathbb{E}_{s_0 \sim \hat{\rho}_0} \mathbb{E}_{a_0 \sim \pi_0^D(\cdot|s_0)} \cdots \mathbb{E}_{s_i \sim T(\cdot|s_{i-1}, a_{i-1}), a_i \sim \pi_0^D(\cdot|s_i)} \right].$$

We are almost done once the $\mathcal{B}^{\pi_0^D}$ in Eq. (9) is replaced by $\mathcal{B}^{\pi}$. Note that

$$\mathbb{E}_{(s,a) \sim \rho_{\hat{\rho}_0}^{\pi_0^D}(s)\pi_0^D(a|s)} \left|Q(s,a) - \mathcal{B}^{\pi_0^D} Q(s,a)\right|$$

$$\leq \mathbb{E}_{(s,a) \sim \rho_{\hat{\rho}_0}^{\pi_0^D}(s)\pi_0^D(a|s)} \left|Q(s,a) - \mathcal{B}^{\pi} Q(s,a)\right| + \left|\mathcal{B}^{\pi} Q(s,a) - \mathcal{B}^{\pi_0^D} Q(s,a)\right|$$

$$= \mathbb{E}_{(s,a) \sim \rho_{\hat{\rho}_0}^{\pi_0^D}(s)\pi_0^D(a|s)} \left|Q(s,a) - \mathcal{B}^{\pi} Q(s,a)\right| + \gamma \left|\mathbb{E}_{s' \sim T(\cdot|s,a)} \mathbb{E}_{\pi, \pi_0^D} Q(s', \pi(s')) - Q(s', \pi_0^D(s'))\right|$$

$$\overset{Lem. \ 2}{\leq} \mathbb{E}_{(s,a) \sim \rho_{\hat{\rho}_0}^{\pi_0^D}(s)\pi_0^D(a|s)} \left[ \left|Q(s,a) - \mathcal{B}^{\pi} Q(s,a)\right| + \gamma L_{\mathcal{A}} \mathbb{E}_{s' \sim T(\cdot|s,a)} W_1(\pi(\cdot|s')||\pi_0^D \cdot |s') \right]$$

$$= \mathbb{E}_{(s,a) \sim \rho_{\hat{\rho}_0}^{\pi_0^D}(s)\pi_0^D(a|s)} \left|Q(s,a) - \mathcal{B}^{\pi} Q(s,a)\right|$$

$$+ L_{\mathcal{A}} \mathbb{E}_{s \sim \rho_{\hat{\rho}_0}^{\pi_0^D}} W_1(\pi(\cdot|s)||\pi_0^D(\cdot|s)) - (1-\gamma) L_{\mathcal{A}} \mathbb{E}_{s \sim \hat{\rho}_0} W_1(\pi(\cdot|s)||\pi_0^D(\cdot|s))$$

$$\leq \mathbb{E}_{(s,a) \sim \rho_{\hat{\rho}_0}^{\pi_0^D}(s)\pi_0^D(a|s)} \left|Q(s,a) - \mathcal{B}^{\pi} Q(s,a)\right| + L_{\mathcal{A}} W_1(\pi(\cdot|s)||\pi_0^D(\cdot|s)). \tag{10}$$

The second-last line follows from that the distribution of $s'$ is $\rho_{\hat{\rho}_0, 1}^{\pi_0^D}(s') = \int_{sa} T(s'|s,a) \rho_{\hat{\rho}_0}^{\pi_0^D}(s,a)$, which satisfies the identity

$$\rho_{\hat{\rho}_0}^{\pi_0^D}(s') = (1-\gamma)\hat{\rho}_0(s') + \gamma \int T(s'|s,a) \rho_{\hat{\rho}_0}^{\pi_0^D}(s,a) ds da = (1-\gamma)\hat{\rho}_0(s') + \gamma \rho_{\hat{\rho}_0, 1}^{\pi_0^D}(s').$$

Combining Eq. (9) and Eq. (10), the result follows. $\qquad \square$

**Lemma 5.** *Let $0 \leq f(s,a) \leq \Delta$ be a bounded function for $(s,a) \in \mathcal{S} \times \mathcal{A}$. Let $\tilde{E}_0^{\infty}$ be the averaging-discounted operator with infinite-length trajectories in $N$ episodes; i.e., $\tilde{E}_0^{\infty} f =$*

$\frac{1}{N} \sum_{e=1}^{N} \sum_{i=0}^{\infty} (1 - \gamma)\gamma^i f(s_i^e, a_i^e)$. Then, with probability greather than $1 - \delta$,

$$\left( \tilde{E}_0^\infty - \mathbb{E}_{\rho_{\hat{\rho}_0}^{\pi_0^D}(s)\pi_0^D(a|s)} \right) f \leq \Delta \sqrt{\frac{2}{N} \log \frac{1}{\delta}}.$$

*Proof.* Let $\delta_{s_0^e}(s)$ be the delta measure at the initial state in episode $e$. Because the empirical distribution $\hat{\rho}_0$ is composed of the realization of the trajectories's initial states, we know $\hat{\rho}_0 = \frac{1}{N} \sum_{e=1}^{N} \delta_{s_0^e}$. Then, Lemma 1 implies $\rho_{\hat{\rho}_0}^{\pi_0^D}$ is an average of the normalized occupancy measures in $N$ episodes.

$$\rho_{\hat{\rho}_0}^{\pi_0^D}(s)\pi_0^D(a|s) \overset{a.e.}{=} \frac{1}{N} \sum_{e=1}^{N} \rho_{\delta_{s_0^e}}^{\pi^e}(s)\pi^e(a|s)$$

$$\overset{def. \ of \ \rho_{\delta_{s_0^e}}^{\pi^e}}{=} \frac{1}{N} \sum_{e=1}^{N} \sum_{i=0}^{\infty} (1 - \gamma)\gamma^i \rho_i^e(s|\delta_{s_0^e}, \pi^e, T)\pi^e(a|s), \tag{11}$$

where $\rho_i^e$ is the state density at trajectory step $i$ in episode $e$. Since $(s_i^e, a_i^e) \sim \rho_i^e(s|\delta_{s_0^e}, \pi^e, T)\pi^e(a|s)$, we have

$$\left( \tilde{E}_0^\infty - \mathbb{E}_{\rho_{\hat{\rho}_0}^{\pi_0^D}(s)\pi_0^D(a|s)} \right) f = \frac{1}{N} \sum_{e=1}^{N} \left[ \sum_{i=0}^{\infty} (1 - \gamma)\gamma^i f(s_i^e, a_i^e) - \mathbb{E}_{\rho_{\delta_{s_0^e}}^{\pi^e}(s)\pi^e(a|s)} f \right]$$

$$\overset{(11)}{=} \frac{1}{N} \sum_{e=1}^{N} \left[ \sum_{i=0}^{\infty} (1 - \gamma)\gamma^i \left[ f(s_i^e, a_i^e) - \mathbb{E}[f(s_i^e, a_i^e)|s_0^e] \right] \right] = \frac{1}{N} \sum_{e=1}^{N} M^e. \tag{12}$$

We claim that $\{M^e\}_{e=1}^{N}$ defined as $M^e = \sum_{i=0}^{\infty} (1 - \gamma)\gamma^i \left[ f(s_i^e, a_i^e) - \mathbb{E}[f(s_i^e, a_i^e)|s_0^e] \right]$ is a martingale difference sequence. To see this, let $\mathcal{F}^e$ be the filtration of all randomness from episode 1 to $e$, with $\mathcal{F}^0$ being a null set. Clearly, we have $M^e \in \mathcal{F}^e$, $\mathbb{E}[M^e|\mathcal{F}^{e-1}] = 0$ and $M^e \in [-\Delta, \Delta]$ since $f(s, a) \in [0, \Delta]$ by assumption, which proves $\{M^e\}_{e=1}^{N}$ is a martingale difference sequence.

Finally, since $M^e$ is bounded in $[-\Delta, \Delta]$, by Azuma-Hoeffding inequality, we conclude that with probability greater than $1 - \delta$,

$$\text{Eq. (12)} \leq N^{-1} \sqrt{\frac{\sum_{e=1}^{N}(2\Delta)^2}{2} \log \frac{1}{\delta}} = \Delta \sqrt{\frac{2}{N} \log \frac{1}{\delta}}.$$

$\square$

**Theorem 1.** *Let $N$ be the number of episodes. For $f: \mathcal{S} \times \mathcal{A} \to \mathbb{R}$, define the operator $\tilde{E}_0^\infty f \triangleq \frac{1}{N} \sum_{e=1}^{N} \sum_{i=0}^{\infty} (1 - \gamma)\gamma^i f(s_i^e, a_i^e)$. Denote the policy mismatch error and the Bellman error as $W_1^{0,\infty} = \tilde{E}_0^\infty W_1(\pi^N||\pi_0^D)(\cdot)$ and $\epsilon_{\hat{Q}}^{0,\infty} = \tilde{E}_0^\infty |\hat{Q}(\cdot, \cdot) - (\mathcal{B}^{\pi^N}\hat{Q})(\cdot, \cdot)|$, respectively. Assume*

*(1) For each $s \in \mathcal{S}$, $\hat{Q}(s, a)$ and $Q^{\pi^N}(s, a)$ are $L_\mathcal{A}$-Lipschitz in $a$.*

*(2) $\hat{Q}(s, a)$ and $Q^{\pi^N}(s, a)$ are bounded and take values in $[0, r^{\max}/(1 - \gamma)]$.*

*(3) The action space is bounded with diameter $diam_\mathcal{A} < \infty$.*

*Note $W_1(\pi^N||\pi_i^D)(s)$ means $W_1(\pi^N(\cdot|s)||\pi_i^D(\cdot|s))$, which is a function of $s$. Then, with probability greater than $1 - \delta$, we have*

$$\mathbb{E}_{(s_0,a_0)\sim\rho_0(s)\pi_0^D(a|s)} \left| \hat{Q}(s_0, a_0) - Q^{\pi^N}(s_0, a_0) \right|$$

$$\leq \frac{r^{\max}}{1 - \gamma} \sqrt{\frac{1}{2N} \log \frac{2}{\delta}} + \left( \frac{r^{\max}}{(1 - \gamma)^2} + \frac{2L_\mathcal{A} diam_\mathcal{A}}{1 - \gamma} \right) \sqrt{\frac{2}{N} \log \frac{2}{\delta}} + \frac{1}{1 - \gamma} \left( \epsilon_{\hat{Q}}^{0,\infty} + 2L_\mathcal{A} W_1^{0,\infty} \right).$$

*Proof.* The proof basically combines Lemma 3, 4 and 5. To start with, the objective is decomposed as:

$$\mathbb{E}_{(s_0,a_0)\sim\rho_0(s)\pi_0^D(a|s)}\left|\hat{Q}(s_0,a_0)-Q^{\pi^N}(s_0,a_0)\right|$$

$$=H_1+\mathbb{E}_{(s_0,a_0)\sim\hat{\rho}_0(s)\pi_0^D(a|s)}\left|\hat{Q}(s_0,a_0)-Q^{\pi^N}(s_0,a_0)\right|$$

$$\leq H_1+\mathbb{E}_{(s,a)\sim\hat{\rho}_0(s)\pi_0^D(a|s)}\left|\hat{Q}(s_0,a_0)-Q^{\pi_0^D}(s_0,a_0)\right|+\left|Q^{\pi_0^D}(s_0,a_0)-Q^{\pi^N}(s_0,a_0)\right|$$

$$\overset{4,\,3}{\leq}H_1+\frac{1}{1-\gamma}\mathbb{E}_{(s,a)\sim\rho_{\hat{\rho}_0}^{\pi_0^D}(s)\pi_0^D(a|s)}\left[\left|\hat{Q}(s,a)-(\mathcal{B}^{\pi^N}\hat{Q})(s,a)\right|+2L_{\mathcal{A}}W_1(\pi^N(\cdot|s)||\pi_0^D(\cdot|s))\right]$$

$$=H_1+\frac{H_2}{1-\gamma}+\frac{1}{1-\gamma}\tilde{E}_0^\infty\left[\left|\hat{Q}(\cdot,\cdot)-(\mathcal{B}^{\pi^N}\hat{Q})(\cdot,\cdot)\right|+2L_{\mathcal{A}}W_1(\pi^N||\pi_0^D)(\cdot)\right]$$

$$\overset{\text{Assump.}}{\leq}H_1+\frac{H_2}{1-\gamma}+\frac{1}{1-\gamma}\left(\epsilon_{\hat{Q}}^{0,\infty}+2L_{\mathcal{A}}W_1^{0,\infty}\right).$$

$$(13)$$

Because $|\hat{Q}(s,a)-Q^{\pi^N}(s,a)|\in[0,\frac{r^{\max}}{1-\gamma}]$, we know $\mathbb{E}_{a\sim\pi_0^D(\cdot|s)}|\hat{Q}(s,a)-Q^{\pi^N}(s,a)|\in[0,\frac{r^{\max}}{1-\gamma}]$, too. Suppose $\hat{\rho}_0$ have $N$ independent samples. By Hoeffding's inequality, with probability $\geq 1-\delta$,

$$H_1=\left(\mathbb{E}_{s_0\sim\rho_0}-\mathbb{E}_{s_0\sim\hat{\rho}_0}\right)\mathbb{E}_{a_0\sim\pi_0^D(\cdot|s_0)}|\hat{Q}(s_0,a_0)-Q^{\pi^N}(s_0,a_0)|\leq\frac{r^{\max}}{1-\gamma}\sqrt{\frac{1}{2N}\log\frac{1}{\delta}}. \quad (14)$$

Also, let $f(s,a)=\left|\hat{Q}(s,a)-(\mathcal{B}^{\pi^N}\hat{Q})(s,a)\right|+2L_{\mathcal{A}}W_1(\pi^N(\cdot|s)||\pi_0^D(\cdot|s))$. Then $f(s,a)$ is bounded in $[0,\frac{r^{\max}}{1-\gamma}+2L_{\mathcal{A}}\text{diam}_{\mathcal{A}}]$. Thereby, Lemma 5 implies that with probability greater than $1-\delta$,

$$H_2=\left(\mathbb{E}_{(s,a)\sim\rho_{\hat{\rho}_0}^{\pi_0^D}(s)\pi_0^D(a|s)}-\tilde{E}_0^\infty\right)\left[\left|\hat{Q}(s,a)-(\mathcal{B}^{\pi^N}\hat{Q})(s,a)\right|+2L_{\mathcal{A}}W_1(\pi^N(\cdot|s)||\pi_0^D(\cdot|s))\right]$$

$$\leq\left(\frac{r^{\max}}{1-\gamma}+2L_{\mathcal{A}}\text{diam}_{\mathcal{A}}\right)\sqrt{\frac{2}{N}\log\frac{1}{\delta}}$$

$$(15)$$

Combining Eq. (13), (14) and (15), a union bound implies with probability greater than $1-2\delta$,

$$\mathbb{E}_{(s_0,a_0)\sim\rho_0(s)\pi_0^D(a|s)}\left|\hat{Q}(s_0,a_0)-Q^{\pi^N}(s_0,a_0)\right|$$

$$\leq\frac{r^{\max}}{1-\gamma}\sqrt{\frac{1}{2N}\log\frac{1}{\delta}}+\left(\frac{r^{\max}}{(1-\gamma)^2}+\frac{2L_{\mathcal{A}}\text{diam}_{\mathcal{A}}}{1-\gamma}\right)\sqrt{\frac{2}{N}\log\frac{1}{\delta}}+\frac{1}{1-\gamma}\left(\epsilon_{\hat{Q}}^{0,\infty}+2L_{\mathcal{A}}W_1^{0,\infty}\right)$$

Finally, rescaling $\delta$ to $\delta/2$ finishes the proof. $\square$

**Corollary 1.** *Let $\tilde{E}_i^L$ be the operator defined as $\tilde{E}_i^L f=\frac{1}{N}\sum_{e=1}^N\sum_{j=i}^{L-1}(1-\gamma)\gamma^{j-i}f(s_j^e,a_j^e)$. Fix assumptions (1) (2) (3) of Theorem 1. Rewrite the policy mismatch error and the Bellman error as $W_1^{i,L}=\tilde{E}_i^L W_1(\pi^N||\pi_i^D)(\cdot)$ and $\epsilon_{\hat{Q}}^{i,L}=\tilde{E}_i^L|\hat{Q}(\cdot,\cdot)-(\mathcal{B}^{\pi^N}\hat{Q})(\cdot,\cdot)|$, respectively, where $W_1(\pi^N||\pi_i^D)(s)=W_1(\pi^N(\cdot|s)||\pi_i^D(\cdot|s))$. Let $\overline{\rho}_i(s)$ be the average state density at trajectory step $i$ over all $N$ episodes. Then, with probability greater than $1-\delta$, we have*

$$\mathbb{E}_{(s_i,a_i)\sim\overline{\rho}_i(s)\pi_i^D(a|s)}\left|\hat{Q}(s_i,a_i)-Q^{\pi^N}(s_i,a_i)\right|\leq$$

$$\frac{r^{\max}}{1-\gamma}\sqrt{\frac{1}{2N}\log\frac{2}{\delta}}+\left(\frac{r^{\max}}{(1-\gamma)^2}+\frac{2L_{\mathcal{A}}diam_{\mathcal{A}}}{1-\gamma}\right)\left(\sqrt{\frac{2}{N}\log\frac{2}{\delta}}+\gamma^{L-i}\right)+\frac{1}{1-\gamma}\left(\epsilon_{\hat{Q}}^{i,L}+2L_{\mathcal{A}}W_1^{i,L}\right).$$

*Moreover, if $i\leq L-\frac{\log\epsilon}{\log\gamma}$ and the constants are normalized as $L_{\mathcal{A}}=c/(1-\gamma)$, $\epsilon_{\hat{Q}}^{i,L}=\xi_{\hat{Q}}^{i,L}/(1-\gamma)$, then, with probability greater than $1-\delta$, we have*

$$\mathbb{E}_{(s_i,a_i)\sim\overline{\rho}_i(s)\pi_i^D(a|s)}\left|\hat{Q}(s_i,a_i)-Q^{\pi^N}(s_i,a_i)\right|$$

$$\leq\tilde{O}\left(\frac{1}{(1-\gamma)^2}\left((r^{\max}+c\cdot diam_{\mathcal{A}})(1/\sqrt{N}+\epsilon)+\xi_{\hat{Q}}^{i,L}+c\cdot W_1^{i,L}\right)\right)$$

*Proof.* Notice that Theorem 1 is the situation when $i = 0$ and $L = \infty$. To push this to $i \geq 0$ and $L = \infty$, recall that Lemma 1 defines the behavior policy $\pi_i^D$, the average state distribution $\bar{\rho}_i$ and the state occupancy measure $\rho_{\bar{\rho}_i}^{\pi_i^D}$ at step $i$. Also, since the trajectories in each episode are initialized using the same initial state distribution, we have $\rho_0 = \bar{\rho}_0$. Therefore, the objective of Theorem 1 is actually $\mathbb{E}_{(s_0,a_0)\sim\bar{\rho}_0(s)\pi_0^D(a|s)}\left|\hat{Q}(s_0,a_0) - Q^{\pi^N}(s_0,a_0)\right|$. This is generalized to $i \geq 0$ using substitutions: $(\bar{\rho}_0, \pi_0^D, \tilde{E}_0^\infty) \to (\bar{\rho}_i, \pi_i^D, \tilde{E}_i^\infty)$, yielding

$$\mathbb{E}_{(s_i,a_i)\sim\bar{\rho}_i(s)\pi_i^D(a|s)}\left|\hat{Q}(s_i,a_i) - Q^{\pi^N}(s_i,a_i)\right|$$
$$\leq \frac{r^{\max}}{1-\gamma}\sqrt{\frac{1}{2N}\log\frac{2}{\delta}} + \left(\frac{r^{\max}}{(1-\gamma)^2} + \frac{2L_\mathcal{A}\mathrm{diam}_\mathcal{A}}{1-\gamma}\right)\sqrt{\frac{2}{N}\log\frac{2}{\delta}} + \frac{1}{1-\gamma}\left(\epsilon_{\hat{Q}}^{i,\infty} + 2L_\mathcal{A}W_1^{i,\infty}\right).$$

Finally, observe that for any bounded $f$:

$$\tilde{E}_i^\infty f \leq \tilde{E}_i^L f + \gamma^{L-i}\|f\|_\infty$$

We arrive at

$$\mathbb{E}_{(s_i,a_i)\sim\bar{\rho}_i(s)\pi_i^D(a|s)}\left|\hat{Q}(s_i,a_i) - Q^{\pi^N}(s_i,a_i)\right|$$
$$\leq \frac{r^{\max}}{1-\gamma}\sqrt{\frac{1}{2N}\log\frac{2}{\delta}} + \left(\frac{r^{\max}}{(1-\gamma)^2} + \frac{2L_\mathcal{A}\mathrm{diam}_\mathcal{A}}{1-\gamma}\right)\left(\sqrt{\frac{2}{N}\log\frac{2}{\delta}} + \gamma^{L-i}\right) + \frac{1}{1-\gamma}\left(\epsilon_{\hat{Q}}^{i,L} + 2L_\mathcal{A}W_1^{i,L}\right).$$

As for the second conclusion, with the substitutions in the statement, we know $\gamma^{L-i} \leq \epsilon$. Hence,

$$\mathbb{E}_{(s_i,a_i)\sim\bar{\rho}_i(s)\pi_i^D(a|s)}\left|\hat{Q}(s_i,a_i) - Q^{\pi^N}(s_i,a_i)\right|$$
$$\leq \frac{r^{\max}}{1-\gamma}\sqrt{\frac{1}{2N}\log\frac{2}{\delta}} + \left(\frac{r^{\max}}{(1-\gamma)^2} + \frac{2c\cdot\mathrm{diam}_\mathcal{A}}{(1-\gamma)^2}\right)\left(\sqrt{\frac{2}{N}\log\frac{2}{\delta}} + \epsilon\right) + \frac{1}{(1-\gamma)^2}\left(\xi_{\hat{Q}}^{i,L} + 2c\cdot W_1^{i,L}\right)$$
$$= \tilde{O}\left(\frac{1}{(1-\gamma)^2}\left((r^{\max} + c\cdot\mathrm{diam}_\mathcal{A})(1/\sqrt{N} + \epsilon) + \xi_{\hat{Q}}^{i,L} + c\cdot W_1^{i,L}\right)\right)$$

$\square$

## A.5 POLICY EVALUATION ERROR UNDER NON-UNIFORM WEIGHTS

**Lemma 6.** *Following from Lemma 1, let* $w = \{w_e > 0\}_{e=1}^N$ *be the (unnormalized) weights for the episodes. We have for the weighted case,* $\frac{1}{\sum_e w_e}\sum_e w_e\rho_{\rho_i^e}^{\pi^e}(s) = \rho_{\bar{\rho}_{i,w}}^{\pi_{i,w}^D}(s)$ *a.e., where* $\rho_{\bar{\rho}_{i,w}}^{\pi_{i,w}^D}(s)$ *is the normalized state occupancy measure is generated by* $(\bar{\rho}_{i,w}, \pi_{i,w}^D, T)$. *Also,* $\frac{1}{\sum_e w_e}\sum_e w_e\rho_{\rho_i^e}^{\pi^e}(s)\pi^e(a|s) = \rho_{\bar{\rho}_{i,w}}^{\pi_{i,w}^D}(s)\pi_{i,w}^D(a|s)$ *a.e.*

*Proof.* The proof is basically a generalization of Lemma 1. Since $\rho_{\rho_i^e}^{\pi^e}$ is the normalized occupancy measure generated by $(\rho_i^e, \pi^e, T)$, we know each $\rho_{\rho_i^e}^{\pi^e}$ is the fixed-point of the Bellman flow equation:

$$\rho_{\rho_i^e}^{\pi^e}(s) = (1-\gamma)\rho_i^e(s) + \gamma\int T(s|s',a')\pi^e(a'|s')\rho_{\rho_i^e}^{\pi^e}(s')ds'da', \quad \forall e \in [1,...,N].$$

Taking a weighted average, we get

$$\frac{1}{\sum_e w_e} \sum_e w_e \rho^{\pi^e}_{\rho^e_i}(s)$$

$$= \frac{1}{\sum_e w_e} \sum_e w_e \left[ (1-\gamma)\rho^e_i(s) + \gamma \int T(s|s',a')\pi^e(a'|s')\rho^{\pi^e}_{\rho^e_i}(s')ds'da' \right]$$

$$= (1-\gamma)\overline{\rho}_{i,w}(s) + \gamma \int T(s|s',a')\frac{1}{\sum_e w_e} \sum_e w_e \left[ \pi^e(a'|s')\rho^{\pi^e}_{\rho^e_i}(s') \right] ds'da'$$

$$= (1-\gamma)\overline{\rho}_{i,w}(s) + \gamma \int T(s|s',a')\frac{\sum_e w_e \pi^e(a'|s')\rho^{\pi^e}_{\rho^e_i}(s')}{\sum_e w_e \rho^{\pi^e}_{\rho^e_i}(s')}\frac{\sum_e w_e \rho^{\pi^e}_{\rho^e_i}(s')}{\sum_e w_e}ds'da'$$

$$= (1-\gamma)\overline{\rho}_{i,w}(s) + \gamma \int T(s|s',a')\pi^D_{i,w}(a'|s')\frac{1}{\sum_e w_e} \sum_e w_e \rho^{\pi^e}_{\rho^e_i}(s')ds'da',$$

where $\pi^D_i(a|s) \triangleq \frac{\sum_e w_e \pi^e(a|s)\rho^{\pi^e}_{\rho^e_i}(s)}{\sum_e w_e \rho^{\pi^e}_{\rho^e_i}(s)}$ is the weighted behavior policy at step $i$. Since the Bellman flow operator is a $\gamma$-contraction in TV distance and hence has a unique (up to difference in some measure zero set) fixed point, denoted as $\rho^{\pi^D_{i,w}}_{\overline{\rho}_{i,w}}(s)$, we arrive at $\frac{1}{\sum_e w_e} \sum_e w_e \rho^{\pi^e}_{\rho^e_i}(s) = \rho^{\pi^D_{i,w}}_{\overline{\rho}_{i,w}}(s)$ a.e. Finally, by definition of $\pi^D_{i,w}(a|s)$, we conclude that $\frac{1}{\sum_e w_e} \sum_e w_e \rho^{\pi^e}_{\rho^e_i}(s)\pi^e(a|s) = \rho^{\pi^D_{i,w}}_{\overline{\rho}_{i,w}}(s)\pi^D_{i,w}(a|s)$ a.e. $\qquad \square$

**Theorem 2.** *Let $w = \{w_e > 0\}^N_{e=1}$ be the (unnormalized) weights for the episodes. Let $\tilde{E}^\infty_{0,w}$ be the operator defined as $\tilde{E}^\infty_{0,w} f = \frac{1}{\sum_e w_e} \sum_{e=1}^N \sum_{i=0}^\infty (1-\gamma)\gamma^i w_e f(s^e_i, a^e_i)$. Fix assumptions (1) (2) (3) of Theorem 1. Rewrite the policy mismatch error and the Bellman error as $W^{0,\infty}_{1,w} = \tilde{E}^\infty_{0,w} W_1(\pi^N||\pi^D_{0,w})(\cdot)$ and $\epsilon^{0,\infty}_{\hat{Q},w} = \tilde{E}^\infty_{0,w}|\hat{Q}(\cdot,\cdot) - (\mathcal{B}^{\pi^N}\hat{Q})(\cdot,\cdot)|$, respectively, where $W_1(\pi^N||\pi^D_{0,w})(s) = W_1(\pi^N(\cdot|s)||\pi^D_{0,w}(\cdot|s))$. With probability greater than $1-\delta$, we have*

$$\mathbb{E}_{(s_0,a_0)\sim\rho_0(s)\pi^D_{0,w}(a|s)}\left|\hat{Q}(s_0,a_0) - Q^{\pi^N}(s_0,a_0)\right| \le \frac{r^{\max}}{1-\gamma}\sqrt{\frac{\sum_e w^2_e}{2(\sum_e w_e)^2}\log\frac{2}{\delta}}$$

$$+ \left(\frac{r^{\max}}{(1-\gamma)^2} + \frac{2L_\mathcal{A}diam_\mathcal{A}}{1-\gamma}\right)\sqrt{\frac{2\sum_e w^2_e}{(\sum_e w_e)^2}\log\frac{2}{\delta}} + \frac{1}{1-\gamma}\left(\epsilon^{0,\infty}_{\hat{Q},w} + 2L_\mathcal{A}W^{0,\infty}_{1,w}\right).$$

*Proof.* The proof basically follows from Theorem 1. Decompose the objective using Lemma 3, 4 as:

$$\mathbb{E}_{(s_0,a_0)\sim\rho_0(s)\pi^D_{0,w}(a|s)}\left|\hat{Q}(s_0,a_0) - Q^{\pi^N}(s_0,a_0)\right|$$

$$= H_1 + \mathbb{E}_{(s_0,a_0)\sim\hat{\rho}_{0,w}(s)\pi^D_{0,w}(a|s)}\left|\hat{Q}(s_0,a_0) - Q^{\pi^N}(s_0,a_0)\right|$$

$$\le H_1 + \mathbb{E}_{(s,a)\sim\hat{\rho}_{0,w}(s)\pi^D_{0,w}(a|s)}\left|\hat{Q}(s_0,a_0) - Q^{\pi^D_{0,w}}(s_0,a_0)\right| + \left|Q^{\pi^D_{0,w}}(s_0,a_0) - Q^{\pi^N}(s_0,a_0)\right|$$

$$\overset{4,3}{\le} H_1 + \frac{1}{1-\gamma}\mathbb{E}_{(s,a)\sim\rho^{\pi^D_{0,w}}_{\hat{\rho}_{0,w}}(s)\pi^D_{0,w}(a|s)}\left[\left|\hat{Q}(s,a) - (\mathcal{B}^{\pi^N}\hat{Q})(s,a)\right| + 2L_\mathcal{A}W_1(\pi^N(\cdot|s)||\pi^D_{0,w}(\cdot|s))\right]$$

$$= H_1 + \frac{H_2}{1-\gamma} + \frac{1}{1-\gamma}\tilde{E}^\infty_{0,w}\left[\left|\hat{Q}(\cdot,\cdot) - (\mathcal{B}^{\pi^N}\hat{Q})(\cdot,\cdot)\right| + 2L_\mathcal{A}W_1(\pi^N||\pi^D_{0,w})(\cdot)\right]$$

$$\overset{\text{Assump.}}{\le} H_1 + \frac{H_2}{1-\gamma} + \frac{1}{1-\gamma}\left(\epsilon^{0,\infty}_{\hat{Q},w} + 2L_\mathcal{A}W^{0,\infty}_{1,w}\right),$$

$$(16)$$

where $\hat{\rho}_{0,w}(s) = \frac{\sum_e w_e \delta(s-s^e_0)}{\sum_e w_e}$ is the weighted empirical initial state distribution. Because $|\hat{Q}(s,a) - Q^{\pi^N}(s,a)| \in [0, \frac{r^{\max}}{1-\gamma}]$, we know $\mathbb{E}_{a\sim\pi^D_{0,w}(\cdot|s)}|\hat{Q}(s,a) - Q^{\pi^N}(s,a)| \in [0, \frac{r^{\max}}{1-\gamma}]$, too.

By Hoeffding's inequality, with probability greater than $1 - \delta$,

$$H_1 = \left(\mathbb{E}_{s_0 \sim \rho_0} - \mathbb{E}_{s_0 \sim \hat{\rho}_{0,w}}\right)\mathbb{E}_{a_0 \sim \pi_{0,w}^D(\cdot|s_0)}\left|\hat{Q}(s_0, a_0) - Q^{\pi^N}(s_0, a_0)\right|$$

$$\leq \frac{1}{\sum_e w_e}\sqrt{\frac{\sum_e \left(w_e \frac{r^{\max}}{1-\gamma}\right)^2}{2}\log\frac{1}{\delta}} = \frac{r^{\max}}{1-\gamma}\sqrt{\frac{\sum_e w_e^2}{2(\sum_e w_e)^2}\log\frac{1}{\delta}}. \tag{17}$$

Also, let $f(s,a) = \left|\hat{Q}(s,a) - (\mathcal{B}^{\pi^N}\hat{Q})(s,a)\right| + 2L_{\mathcal{A}}W_1(\pi^N(\cdot|s)||\pi_{0,w}^D(\cdot|s))$. Then $f(s,a)$ is bounded in $[0, \frac{r^{\max}}{1-\gamma} + 2L_{\mathcal{A}}\text{diam}_{\mathcal{A}}]$. Using a weighted version of Azuma-Hoeffding in Lemma 5, we have that with probability greater than $1 - \delta$,

$$H_2 = \left(\mathbb{E}_{(s,a)\sim\rho_{\hat{\rho}_{0,w}}^{\pi_{0,w}^D}(s)\pi_{0,w}^D(a|s)} - \tilde{E}_{0,w}^\infty\right)\left[\left|\hat{Q}(s,a) - (\mathcal{B}^{\pi^N}\hat{Q})(s,a)\right| + 2L_{\mathcal{A}}W_1(\pi^N(\cdot|s)||\pi_{0,w}^D(\cdot|s))\right]$$

$$\leq \left(\frac{r^{\max}}{1-\gamma} + 2L_{\mathcal{A}}\text{diam}_{\mathcal{A}}\right)\sqrt{\frac{2\sum_e w_e^2}{(\sum_e w_e)^2}\log\frac{1}{\delta}} \tag{18}$$

Combining Eq. (16), (17) and (18), a union bound implies with probability greater than $1 - 2\delta$,

$$\mathbb{E}_{(s_0,a_0)\sim\rho_0(s)\pi_{0,w}^D(a|s)}\left|\hat{Q}(s_0, a_0) - Q^{\pi^N}(s_0, a_0)\right| \leq \frac{r^{\max}}{1-\gamma}\sqrt{\frac{\sum_e w_e^2}{2(\sum_e w_e)^2}\log\frac{1}{\delta}}$$

$$+ \left(\frac{r^{\max}}{(1-\gamma)^2} + \frac{2L_{\mathcal{A}}\text{diam}_{\mathcal{A}}}{1-\gamma}\right)\sqrt{\frac{2\sum_e w_e^2}{(\sum_e w_e)^2}\log\frac{1}{\delta}} + \frac{1}{1-\gamma}\left(\epsilon_{\hat{Q},w}^{0,\infty} + 2L_{\mathcal{A}}W_{1,w}^{0,\infty}\right)$$

Finally, rescaling $\delta$ to $\delta/2$ finishes the proof. $\qquad\square$

**Corollary 2.** *Let $\tilde{E}_{i,w}^L$ be the operator defined as $\tilde{E}_{i,w}^L f = \frac{1}{\sum_e w_e}\sum_e\sum_{j=i}^{L-1}(1 - \gamma)\gamma^{j-i}w_e f(s_j^e, a_j^e)$. Fix assumptions (1)(2)(3) of Theorem 1. Rewrite the policy mismatch error and the Bellman error as $W_{1,w}^{i,L} = \tilde{E}_{i,w}^L W_1(\pi^N||\pi_{i,w}^D)(\cdot)$ and $\epsilon_{\hat{Q},w}^{i,L} = \tilde{E}_{i,w}^L|\hat{Q}(\cdot,\cdot) - (\mathcal{B}^{\pi^N}\hat{Q})(\cdot,\cdot)|$, respectively, where $W_1(\pi^N||\pi_{i,w}^D)(s) = W_1(\pi^N(\cdot|s)||\pi_{i,w}^D(\cdot|s))$. Let $\bar{\rho}_{i,w}(s)$ be the average weighted state density at trajectory step $i$. Then, with probability greater than $1 - \delta$, we have*

$$\mathbb{E}_{(s_i,a_i)\sim\bar{\rho}_{i,w}(s)\pi_{i,w}^D(a|s)}\left|\hat{Q}(s_i, a_i) - Q^{\pi^N}(s_i, a_i)\right| \leq \frac{r^{\max}}{1-\gamma}\sqrt{\frac{\sum_e w_e^2}{2(\sum_e w_e)^2}\log\frac{2}{\delta}}$$

$$+ \left(\frac{r^{\max}}{(1-\gamma)^2} + \frac{2L_{\mathcal{A}}\text{diam}_{\mathcal{A}}}{1-\gamma}\right)\left(\sqrt{\frac{2\sum_e w_e^2}{(\sum_e w_e)^2}\log\frac{2}{\delta}} + \gamma^{L-i}\right) + \frac{1}{1-\gamma}\left(\epsilon_{\hat{Q},w}^{i,L} + 2L_{\mathcal{A}}W_{1,w}^{i,L}\right).$$

*Moreover, if $i \leq L - \frac{\log\epsilon}{\log\gamma}$ and the constants are normalized as $L_{\mathcal{A}} = c/(1-\gamma)$, $\epsilon_{\hat{Q},w}^{i,L} = \xi_{\hat{Q},w}^{i,L}/(1-\gamma)$, then, with probability greater than $1 - \delta$, we have*

$$\mathbb{E}_{(s_i,a_i)\sim\bar{\rho}_{i,w}(s)\pi_{i,w}^D(a|s)}\left|\hat{Q}(s_i, a_i) - Q^{\pi^N}(s_i, a_i)\right|$$

$$\leq \tilde{O}\left(\frac{1}{(1-\gamma)^2}\left((r^{\max} + c \cdot diam_{\mathcal{A}})\left(\sqrt{(\sum_e w_e^2)/(\sum_e w_e)^2} + \epsilon\right) + \xi_{\hat{Q},w}^{i,L} + c \cdot W_{1,w}^{i,L}\right)\right)$$

*Proof.* Observe that for any bounded $f$:

$$\tilde{E}_{i,w}^\infty f \leq \tilde{E}_{i,w}^L f + \gamma^{L-i}\|f\|_\infty$$

Thus, we can start from Theorem 2 and prove with the same argument in Corollary 1. $\qquad\square$

A.6   PROOF OF EMPHAZING RECENT EXPERIENCE

**Proposition 1.** *The ERE strategy in Wang & Ross (2019) is equivalent to a non-uniform sampling with weight $w_t$:*

$$w_t \propto \frac{1}{1 - \eta^{L_0/K}} \left( \frac{1}{\max(t, c_{\min}, N_0 \eta^{L_0})} - \frac{1}{N_0} \right) + \frac{\mathbb{1}(t \le c_{\min}) K}{c_{\min}} \max\left( 1 - \frac{\ln c_{\min}/N_0}{L_0 \ln \eta}, 0 \right),$$

*where $t$ is the age of a data point relative to the newest time step; i.e., $w_0$ is the newest sample. $N_0$ is the size of the experience replay. $L_0 = 1000$ is the maximum horizon of the environment. $K$ is the length of the recent trajectory. $\eta \approx 0.996$ is the decay parameter. $c_{\min} \approx 5000$ is the minimum coverage of the sampling. Moreover, the ERE strategy can be approximated (by Taylor Approximation) as*

$$w_t \propto \frac{1}{\max(t, c_{\min}, N_0 \eta^{L_0})} - \frac{1}{N_0} + \frac{\mathbb{1}(t \le c_{\min})}{c_{\min}} \max\left( \ln \frac{c_{\min}}{N_0 \eta^{L_0}}, 0 \right)$$

*Proof.* Recall that Wang & Ross (2019) assume a situation of doing $K$ updates in each episode. In the $k$th update, the data is sampled uniformly from the most recent $c_k = \max(N_0 \eta^{k \frac{L_0}{K}}, c_{\min})$ points.

To compute the aggregated weight $w_t$ over these $K$ updates, observe that a data point of age $t$ is in the most recent $c_k$ points if $c_k \ge t$ and that the weight in each uniform sample is $1/c_k$. Therefore, $w_t$ should be proportional to

$$w_t \propto \sum_{\substack{k:\ 1 \le k \le K, \\ c_k \ge t}} \frac{1}{c_k}. \tag{19}$$

Because $c_k$ is designed to be lower bounded by $c_{\min}$, we shall discuss Eq. (19) by cases.

(1) When $t > c_{\min}$, we know $c_k > c_{\min}$ because $c_k \ge t$ is a constraint in the sum. This means $c_k = N_0 \eta^{k L_0/K}$ and hence $c_k \ge t$ is equivalent to $k \le \ln \frac{t}{N_0} / \ln \eta^{L_0/K}$. Eq. (19) becomes

$$\sum_{\substack{k:\ 1 \le k \le K, \\ c_k \ge t}} \frac{1}{c_k} = \sum_{k=1}^{\min(K,\ \ln \frac{t}{N_0} / \ln \eta^{L_0/K})} \frac{1}{N_0} \eta^{-L_0 k/K} \overset{(*)}{=} \sum_{k=1}^{\min(K,\ \ln \frac{t}{N_0} / \ln \xi)} \frac{1}{N_0} \xi^{-k}$$

$$= \begin{cases} \frac{\xi^{-1}}{N_0} \frac{1 - \xi^{-\log_\xi \frac{t}{N_0}}}{1 - \xi^{-1}} = \frac{1/t - 1/N_0}{1 - \eta^{L_0/K}} & \text{if } K > \ln \frac{t}{N_0} / \ln \eta^{L_0/K} \\ \frac{\xi^{-1}}{N_0} \frac{1 - \xi^{-K}}{1 - \xi^{-1}} = \frac{\eta^{-L_0}/N_0 - 1/N_0}{1 - \eta^{L_0/K}} & \text{if } K \le \ln \frac{t}{N_0} / \ln \eta^{L_0/K} \end{cases}$$

$$= \begin{cases} \frac{1/t - 1/N_0}{1 - \eta^{L_0/K}} & \text{if } t > N_0 \eta^{L_0} \\ \frac{1/(N_0 \eta^{L_0}) - 1/N_0}{1 - \eta^{L_0/K}} & \text{if } t \le N_0 \eta^{L_0} \end{cases} = \frac{1}{1 - \eta^{L_0/K}} \left( \frac{1}{\max(t,\ N_0 \eta^{L_0})} - \frac{1}{N_0} \right)$$

where (*) is a substitution: $\xi = \eta^{L_0/K}$.

(2) When $t \le c_{\min}$, we know $c_k \ge t$ for all $k$ because $c_k \ge c_{\min}$ by definition. Eq. (19) becomes

$$\sum_{\substack{k:\ 1 \le k \le K, \\ c_k \ge t}} \frac{1}{c_k} = \sum_{k=1}^{K} \frac{1}{c_k}$$

$$\overset{(*)}{=} \sum_{k=1}^{\min(K,\ \ln \frac{c_{\min}}{N_0} / \ln \xi)} \frac{1}{N_0} \xi^{-k} + \max(K - \ln \frac{c_{\min}}{N_0} / \ln \xi,\ 0) \frac{1}{c_{\min}}$$

$$\overset{(**)}{=} \frac{1}{1 - \eta^{L_0/K}} \left( \frac{1}{\max(c_{\min},\ N_0 \eta^{L_0})} - \frac{1}{N_0} \right) + \max(K - K \frac{\ln c_{\min}/N_0}{L_0 \ln \eta},\ 0) \frac{1}{c_{\min}}$$

$$= \frac{1}{1 - \eta^{L_0/K}} \left( \frac{1}{\max(c_{\min},\ N_0 \eta^{L_0})} - \frac{1}{N_0} \right) + \frac{K}{c_{\min}} \max\left( 1 - \frac{\ln c_{\min}/N_0}{L_0 \ln \eta},\ 0 \right),$$

where (*) does a substitution: $\xi = \eta^{L_0/K}$ and split the sum. (**) reuses the analysis in case (1).

Combining cases (1) and (2), we arrive at the first conclusion. As for the approximation, since $\eta \approx 0.996$, let $\eta = 1 - \kappa$. We have

$$1 - \eta^{L_0/K} = 1 - (1 - \kappa)^{L_0/K} \approx 1 - 1 + \kappa L_0/K = \kappa \frac{L_0}{K} \approx -\frac{L_0}{K} \ln(1 - \kappa) = -\frac{L_0}{K} \ln \eta$$

Thus the first term in the conclusion of Prop. 1 is proportional to $K$. The second term is also proportional to $K$. Since $w_t$ is only made to be proportional to the RHS and both terms on the RHS become proportional to $K$, we can remove $K$ on the RHS:

$$w_t \propto \frac{1}{-L_0 \ln \eta} \left( \frac{1}{\max(t, c_{\min}, N_0 \eta^{L_0})} - \frac{1}{N_0} \right) + \frac{\mathbb{1}(t \le c_{\min})}{c_{\min}} \max \left( 1 - \frac{\ln c_{\min}/N_0}{L_0 \ln \eta}, 0 \right)$$

Finally, because $0 < \eta < 1$, $-L_0 \ln \eta$ is a positive number, the above expression can be further simplified by timing $-L_0 \ln \eta$ on the RHS, yielding the result. $\square$

**Proposition 2.** *Let $\{w_t > 0\}_{t=1}^N$ be the weights of the data indexed by $t$. Then the Hoeffding error $\sqrt{\sum_{t=1}^N w_t^2} / \sum_{t=1}^N w_t$ is minimized when the weights are equal: $w_t = c > 0$, $\forall\, t$.*

*Proof.* Let $w = [w_1, ..., w_N]^\top$ be the weight vector and $f(w) = \sqrt{w^\top w}/(\mathbb{1}^\top w)$ be the Hoeffding error. Observe that $f(w)$ is of the form:

$$f(w) = \|w\|/(\mathbb{1}^\top w) = \|w/(\mathbb{1}^\top w)\|,$$

where $\|w\| = \sqrt{w^\top w}$ is the 2-norm of $w$. Thereby, let $z = w/(\mathbb{1}^\top w)$ be the normalized vector. That $f(w)$ is minimized is equivalent to that $g(z) = \|z\|$ is minimized for $\mathbb{1}^\top z = 1$. By the lagrange multiplier, this happens when

$$\nabla g = \frac{z}{\|z\|} = \lambda \mathbb{1}, \text{ for some } \lambda \in \mathbb{R}.$$

This can be achieved by $z_t = c$ for some $c > 0$. Therefore, we know $f$ is minimized when

$$z_t = \frac{w_t}{\mathbb{1}^\top w} = c \text{ for some } c.$$

Since $\mathbb{1}^\top w$ does not depend on $t$, we conclude that the minimizer happens at $w_t = c > 0$, $\forall t$. $\square$

