# OpenReview forum: "Explaining Off-Policy Actor-Critic From A Bias-Variance Perspective"
_ICLR.cc/2022/Conference — ICLR 2022 Submitted_

### Official Review · Reviewer_b574 · 2021-10-27

**Correctness:** 2
**Technical Novelty And Significance:** 2
**Empirical Novelty And Significance:** 2
**Recommendation:** 3
**Confidence:** 3

**Main Review:**

This paper is very confusing.

First, the title is about explaining off-policy actor critic. However, all the theories discuss only the critic error. In my opinion, the authors should focus on the policy evaluation problem with data from mixing distributions because the theory has noting to do with the actor.

Second, Algorithm 1 does not make sense to me. In Line 3, it involves the true Bellman operator, which is not available because we do not know the transition kernel. It also does not specify the function class of Q, so I'll assume the argmin is done over all possible Q function. Then shouldn't the argmin simply return the true Q function of the policy \pi^e, because the true Q function apparently gives 0 for the objective inside the argmin? This confusion makes it hard to interpret Theorem 1. First Theorem 1 does not specify what \hat Q it talks about, I feel it cannot be an arbitrary \hat Q because otherwise the error shouldn't depend on N. For now I assume it's the \hat Q generated by Line 3 in Algorithm 1. Then as discussed before, that argmin should return the true Q function, then the error should simply be 0.

Third, I do not understand the motivation of this work. What do we get from the proposed decomposition? It looks it's just an understanding of ERE, which I don't think is enough for publication. Further, it would be good to explain how many seeds are used to generate the plots. For now it is hard to interpret the results.

**Summary Of The Paper:**

The paper proposes to decompose the critic error into several terms to better understand off-policy actor-critic algorithms. Empirical results are provided to show how their theory can be used to explain two experience replay sampling strategies.

**Summary Of The Review:**

The presentation and motivation of the work is not clear

---

> ### Author Response · Authors · 2021-11-15
> **A response to the questions from Reviewer b574**
>
> We thank Reviewer b574 for the valuable comments. We’ve revised the paper to strengthen the following important points. We hope this helps resolve the general idea and motivation of this work.
>
> (a) Connection from theory to practice
>
> We re-write section 5.2 as “policy evaluation error under non-uniform weights” and stress that Corollary 2’s analysis under non-uniform weights can explain the practical sampling strategies. Essentially, corollary 2 shows that the policy evaluation error is controlled by weighted bias, Hoeffding error (weighted variance), and bellman error. Then, we argue that 1/age-based samplings (ERE, ERE_apx, 1/age) achieve smaller weighted bias and variance than uniform sampling. Thus, they should have a better performance, and we verify this in section 5.3.
>
> (b) Performance comparison with recent work
>
> We provide additional performance comparisons on MuJoCo environments with some recent work (LAP, PAL, Peng’s Q) in Table 1. Still, we find SAC + 1/age-based samplings generally outperform these strong baselines. This suggests 1/age-based samplings not only possess simplicity and theoretical explanations but also achieve better performances.
>
> The detailed comments are addressed below.
>
> Q1: All the theories discuss only the critic error. In my opinion, the authors should focus on the policy evaluation problem with data from mixing distributions because the theory has noting to do with the actor.
>
> A1: We are sorry for the confusion. The theories (Thm1, Cor1, Thm2, Cor2) are policy evaluation errors. This basically quantifies how good $\hat{Q}$ is as an approximator of the current policy’s Q function, $Q^{\pi^N}$. Hence, the theory is tightly related to the actor as the error is defined from the current policy’s perspective.
>
> Q2: Algorithm 1 does not make sense to me.
>
> A2: Thanks for the comments. We find the argmin step not the best description for the algorithm and we’ve changed it to a gradient descent step. After the revision, it is clear that $\hat{Q}$ is not an arbitrary Q function but a neural-network-parameterized Q function optimized by gradient descents.
>
> Q3: I do not understand the motivation of this work. What do we get from the proposed decomposition? It looks it's just an understanding of ERE.
>
> A3: Our theoretical analysis explains the success of “a class of sampling method“, 1/age-based sampling. We don’t simply focus on the ERE. Instead, we show that 1/age-based samplings (ERE, ERE_apx, 1/age) perform almost equally well. Corollary 2 is the formal description for the theoretical explanation and Figure 2 & Table 1 are the experimental results.

---

### Official Review · Reviewer_e5zQ · 2021-10-30

**Correctness:** 2
**Technical Novelty And Significance:** 2
**Empirical Novelty And Significance:** 2
**Recommendation:** 3
**Confidence:** 3

**Details Of Ethics Concerns:**

No concerns.

**Main Review:**

========== Novelty =============

The idea of adjusting sampling probability of transitions in the replay buffer for off-policy learning is not new, and has been explored in a number of orthogonal ways as also mentioned in the current paper. This paper investigates a theoretically motivated approach for decomposing the evaluation errors and studies empirical properties of a number of other sampling methods.

========== Literature reviews =============

As mentioned by the authors, off-policy evaluation learning has been categorized into importance sampling based and regression based. However, in practice, it is more common to combine the aforementioned ideas with contraction operators to carry out scalable off-policy learning when combined with function approximations. Notable examples include n-step Q-learning [1], Retrace [2], Peng's Q [3] among others [4-6]. I think they might be of direct interest if the authors' attempt is to analyze the stability of off-policy learning algorithms.

[1] Hessel et al, Rainbow: combining improvements in deep RL, 2018
[2] Munos et al, Safe and sample efficient RL, 2016
[3] Kozuno et al, Revisiting Peng's Q lambda for modern RL, 2021
[4] Harutyunyan et al, Q(lambda) with off-policy corrections, 2016
[5] Tang et al, Taylor expansion policy optimization, 2020
[6] Rowland et al, Adaptive trade-offs in off-policy learning, 2019

The author has also missed a ref [7] on connecting replay with loss functions.

[7] Fujimoto et al, An equivalence between loss function and non-uniform sampling, 2021

========== Detailed questions =============

1. Overall, the notations are a bit confusing throughout the paper. The paper uses rho_{rho_i} to represent discounted visitation distribution rho starting from rho_i, and uses rho_i to represent the discounted visitation distribution at step i. Putting such notations together makes it quite difficult to read in general. I'd suggest simplifying the notations by e.g., getting rid of a general initial distribution but focusing on a single starting state s_0, or adopting some other ways to represent rho_{rho_i}.

2. The indices e and i are also a bit counterintuitive. I'd suggest using i for the update iteration and t for the time stpe.

3. Algo 1 does not necessarily reflect an exact off-policy actor-critic algorithm used in practice. In particular, the data collection policy might be a perturbed policy (e.g., epsilon-greedy or gaussian corrupted version of the target policy pi^e). I think it is worth making this clear in the presentation.

4. In Eqn 6, an "average" behavior policy is defined using all previous policies. One question I have in mind is the utility of defining such a distribution -- though by combining all discounted visitation distribution, we can define an average policy, this policy does not necessarily induce the average discounted visitation distribution. In other words, there can be an average discounted visitation dist which cannot be realized by any Markovian / non-stationary policy.

5. Based on Thm 1, it seems that the evaluation always suffers from an error that is (1/1-gamma) * W, where W is the wasserstain distance mismatch between the current policy pi^N and average policy pi^D. This observation is based on the last term of the RHS in Thm 1. Does it mean that the error is irreducible when pi^N mismatches with pi^D (i.e., W>0)? I am not quite sure if this is true in the tabular case, in which the evaluation error should vanish as more samples come in -- as long as all state-action pairs are visited, the evaluation error should decrease and vanish. It is not clear how the result in Thm 1 is consistent in this case

6. Though I appreciate that the authors try to connect theory with practice, I find the jump from Sec 4 to Sec 5 to be rather abrupt. In particular, I don't see why ERE's implementation of the sampling scheme directly relates to the error bounds in Sec 4. Can we say ERE seeks to minimize the error bound?

A rather important factor here is that, intuitively, it is kind of clear why putting emphasis on recent experiences makes sense -- this is because overall RL algorithms prefer on-policy data over off-policy data to be more stable. In fact, even algorithms which are off-policy by design can perform much better when using near on-policy data, such as VMPO (over MPO) and R2D2 (over DQN). Therefore, the intuition of using recent experience is already there in the literature. What can be valuable here is how detailed implementation practices seeks to optimize a theoretically justified bound. Unfortunately, it is not reflected well here.

7. Fig 2 shows the difference of different algorithms. It seems that different sampling methods do not have that much of a significant difference in the performance. Overall, the empirical observations do not make a convincing case as to why methods such as ERE matter.


**Summary Of The Paper:**

The paper studies the policy evaluation error in off-policy actor critic algorithms. The core idea is to decompose the error into three terms: bellman error, policy mismatch bias and sampling variance. Based on the decomposition, the paper justifies ERE, a recently proposed replay sampling heuristic and empirically studies the properties of a number of other sampling methods.

**Summary Of The Review:**

Overall, I think the paper is lacking in a few aspects.

1. Theory is not presented in a very clear way. I think the theory results are not very convincing, in that it does not reconcile with certain intuitions. However, I might be missing something here and am curious to hear what the authors say.

2. Theory does not connect with practice. A major point of the paper, to my understanding, is that it entails a potentially theoretically sound framework to explain replay practices. However, I don't think the authors have established a convincing case here -- in understanding the practice, the theory here does not provide much more information than plain intuitions we already have.

---

> ### Author Response · Authors · 2021-11-15
> **A response to the comments from Reviewer e5zQ**
>
> We thank Reviewer e5zQ for pointing out the references that we might have missed. We’ve included them in our literature review and presented an additional performance comparison in Table 1.
>
> The detailed comments are addressed below.
>
> Q1: Overall, the notations are a bit confusing throughout the paper.
>
> A1: We apologize for the confusion. We have improved our notation in the new revision. Currently, our notations are that $\rho_i$ is a state distribution at step i and $\rho_{\rho_i}$ is the occupancy measure generated by $(\rho_i, \pi, T)$. Therefore, $\rho_{\rho_i}$ is an exponential average over $\rho_j$ for $j \in [i, \infty)$.
>
> Another important note is that our behavior policy has a specific construction that requires occupancy measures starting at different distributions. Hence, we probably cannot get rid of general initial distribution and focus on a single starting state.
>
> Q2: The indices e and i are also a bit counterintuitive. I'd suggest using i for the update iteration and t for the time step.
>
> A2: Thanks for the suggestion. e is the episode number and i is the trajectory step in an episode. We adopt this because our analysis depends on the number of episodes and the trajectory step. That is, different trajectory step i has a different error bound, and the variance term decreases in the number of episodes.
>
> Q3: Algo 1 does not necessarily reflect an exact off-policy actor-critic algorithm used in practice. In particular, the data collection policy might be a perturbed policy (e.g., epsilon-greedy or gaussian corrupted version of the target policy $\pi^e$). I think it is worth making this clear in the presentation.
>
> A3: Thanks for pointing out the difference that we might have missed. We have added a discussion on this issue in the new revision. Actually, our analysis framework can address the perturbed policy because our expectations and behavior policies are constructed by data collection policies. Since everything is adaptive to the data collection policies, the perturbation on the policies is not an issue.
>
> Q4: In Eqn 6, an "average" behavior policy is defined using all previous policies. One question I have in mind is the utility of defining such a distribution -- though by combining all discounted visitation distribution, we can define an average policy, this policy does not necessarily induce the average discounted visitation distribution. In other words, there can be an average discounted visitation dist which cannot be realized by any Markovian/non-stationary policy
>
> A4: Thanks for the question. In Lemma 1, we show that our behavior policy can always generate the averaged occupancy measure, and this is also the distribution we work on. Hence, the behavior policy always induces the average discounted visitation distribution, and this is the reason why we construct the behavior policy in our specific way.
>
> Q5: Based on Thm 1, it seems that the evaluation always suffers from an error that is (1/1-gamma) * W, where W is the Wasserstein distance mismatch between the current policy pi^N and average policy pi^D. Does it mean that the error is irreducible when pi^N mismatches with pi^D (i.e., W>0)?
>
> A5: Thanks for the question. Yes, the policy mismatch error is irreducible because we are considering a general MDP in continuous spaces, and it is always possible to find some neighborhoods of (s, a) that are never visited before. As you mention, this is not a problem in a tabular MDP because the space of (s, a) pairs is finite and we can design some exploration to visit all of them. Therefore, because it is impossible to visit all state-action pairs in a continuous space, the policy mismatch error is irreducible (without strong assumptions such as linear MDP).

---

> > ### Author Response · Authors · 2021-11-15
> > **A response to the comments from Reviewer e5zQ (continued)**
> >
> > Q6: Though I appreciate that the authors try to connect theory with practice, I find the jump from Sec 4 to Sec 5 to be rather abrupt. In particular, I don't see why ERE's implementation of the sampling scheme directly relates to the error bounds in Sec 4. Can we say ERE seeks to minimize the error bound?
> >
> > A6: Thanks for the question. We mention at the beginning of sec. 5.2 that we can explain ERE and 1/age using an error analysis under non-uniform aggregate weights in the Appendix (Theorem 2 & Corollary 2). This is a generalization of Thm1 & Cor1 and therefore we put it in the appendix. Nevertheless, the high-level idea is that ERE and 1/age reduce the error bound (by reducing bias & variance) and hence perform better.
> >
> > In other words, our contribution is to first derive a bound on the policy evaluation error and then seek algorithms to minimize it. Our conclusion in a practical sense is that 1/age-based samplings (ERE, ERE_apx, 1/age) perform very similarly and can beat uniform sampling and PER. Later on, we showed that SAC + 1/age-based samplings are enough to beat the prior work (in Table 1). Hence, although using recent experience may not be new, we are the first to analyze it in a theoretical sense and show that a simple change in SAC’s sampling is able to achieve good results. We find such simplicity useful in both practical and theoretical senses.
> >
> > Q7: Fig 2 shows the difference of different algorithms. It seems that different sampling methods do not have that much of a significant difference in the performance. Overall, the empirical observations do not make a convincing case as to why methods such as ERE matter.
> >
> > A7: Figure 2 shows that 1/age-based samplings (ERE, ERE_apx, 1/age) outperform uniform sampling and PER. In the new revision, we provide additional evidence in Table 1, showing that SAC + 1/age-based samplings outperform some recent benchmarks. Hence, we are not just focusing on the ERE but all 1/age-based samplings, and we have explained why they work from a bias-variance perspective.

---

### Official Review · Reviewer_VgPP · 2021-11-02

**Correctness:** 2
**Technical Novelty And Significance:** 2
**Empirical Novelty And Significance:** 2
**Recommendation:** 3
**Confidence:** 5

**Main Review:**

Overall comments :

	1. This paper studies policy evaluation errors for off-policy actor-critic algorithms. The key argument is that the policy evaluation error for actor-critic in off-policy has not been studied well. While this is true, I would have expected some empirical analysis, to justify the theoretical claims of the off-policy critic error in the overall algorithms. For example, how does the critic evaluation error in algorithms like SAC/DDPG lead to the performance in the policy improvement step? Does poor estimation of the critic lead to errors in algorithms like SAC? I think a careful analysis of this is what is expected from a paper trying to analysue bias, variance and policy evaluation error of algorithms like SAC/DDPG etc which has strong empirical performance.

	2. Theorem 1 characterizes the effect of the policy evaluation error in the off-policy actor-critic setting. While this result is important to characterize - I am not sure if it adds too much value in the overall scheme of things. For example, it'd have been better to see how this error leads to the convergence/divergence of off-policy algorithms? Otherwise, this is similar to any Q function error, as in TD learning, with the additional terms due to the policy mismatch term W (but this theorem result is what we'd expect anyway?)

	3. Theorem 1 and the Corollary are useful to see how the overall off-policy learning error can be characterized in terms of the bias and variance. I think this result is important to exactly see the effect policy mismatch and the induced variance has in the overall learning algorithm.

	4. The experimental analysis of this paper is rather weak. The only strong result here is Theorem 1 - but it is difficult to justify the usefulness of it. In the experimental studies, I expected ablation analysis demonstrating the effect of this variance term and how the critic evaluation error influences the performance of standard off-policy actor-critic algorithms like SAC/DDPG.

	5. The paper discusses the different sampling schemes, and I thought the motivation was to see why these sampling schemes often work well in practice. However, from the context of the paper, it is not clear how this contribution is justified - I do not see any clear statements/results backing this claim?


**Summary Of The Paper:**

This paper studies the bias variance perspective of off-policy actor-critic algorithms. The policy mismatch in off-policy learning leads to a variance issue (often due to importance sampling) while the bias is due to the mismatch of the behaviour and the target policies. This paper studies the setting where the behaviour policy is unknown. The paper studies why sampling schemes from the experience replay buffer often work well - since they lead to lower bias and variance issues than uniform sampling from the buffer.  The key contribution of this work is to provide theoretical justification, for off-policy actor-critic algorithms from a bias-variance perspective.


**Summary Of The Review:**

I think this paper is interesting and careful studies of existing algorithms that work well in practice is required. However, this work needs more analysis (either experimentally or theoretical justifications) for it to be ready for acceptance. There is only one theory result (Theorem 1) which characterizes the off-policy actor-critic algorithms. However, this result statement is itself not novel and is derived from a vast majority of similar results in RL literature. In practice, I expected a lot more ablation studies characterizing the bias-variance trade-off. The paper perhaps has more claims than what it could demonstrate - and clearly needs more work for it to be ready for acceptance. I would encourage the authors to pick some simpler tasks and demonstrate the bias/variance analyais on some simple mdps too - instead of only showing performance curves for different sampling schemes on some standard mujoco tasks (which often is hard to interpret and not clear how these results are supporting the advertised claim)

---

> ### Author Response · Authors · 2021-11-15
> **A response to the comments from Reviewer VgPP**
>
> We thank Reviewer VgPP for the valuable comments. We will conduct the additional experiments/study in our future revision. Currently, we’ve revised the paper to strengthen the following points.
>
> 1. Connection from theory to practice
>
> We re-write section 5.2 as “policy evaluation error under non-uniform weights” and stress that Corollary 2’s analysis under non-uniform weights can explain the practical sampling strategies. Essentially, corollary 2 shows that the policy evaluation error is controlled by weighted bias, Hoeffding error (weighted variance), and bellman error. Then, we argue that 1/age-based samplings (ERE, ERE_apx, 1/age) achieve smaller weighted bias and variance than uniform sampling. Thus, they may have a better performance, and we verify this in section 5.3.
>
> 2. Comparison with SAC/DDPG
>
> As mentioned in the second paragraph of sec. 5.3, our experiments are based on SAC. The only difference is that the standard SAC uses uniform sampling from the replay buffer, while we change it to 1/age-based samplings (ERE, ERE_apx, 1/age). Figure 2 shows that SAC + 1/age-based samplings are much better than SAC + uniform/PER. Hence, this shows SAC’s performance can be boosted by simply changing its sampling method.
>
> 3. Performance comparison with recent work
>
> We provide additional performance comparisons on MuJoCo environments with some recent work (LAP, PAL, Peng’s Q) in Table 1. Still, we find SAC + 1/age-based samplings generally outperform these strong baselines. This suggests 1/age-based samplings not only possess simplicity and theoretical explanations but also achieve better performances.

---

### Official Review · Reviewer_RYz4 · 2021-11-04

**Correctness:** 3
**Technical Novelty And Significance:** 4
**Empirical Novelty And Significance:** 3
**Recommendation:** 8
**Confidence:** 4

**Main Review:**

This is a nice paper expanding our undertanding of actor-critic algorithms. Focusing on the policy evaluation error is a simple approach yielding meaningful results. The assumptions are sufficiently weak for the conclusions to apply to modern deep RL methods, a strength of this analysis. The experiments are a good addition to validate the theoretical findings and make the case more convincing. Overall, this is a great paper with concrete insights that are clearly conveyed.

Questions and comments:
- I think the fact that the analysis is uisng continuous action spaces is neat. This seems to be a less popular choice in RL.

- Algorithm 1 is set up nicely to avoid the complexity of deep RL algorithms but still retain the essence of the algorithms and allowing interesting theoretical derivations.

- p.6 To clarify, why is it stated that $|\hat{Q} - Q^{\pi^N}|$  mainly depends on steps $\ge i$. Is this because of the presence of $\epsilon^{i,L}_\hat{Q}$ and $W^{i,L}_1$?

- It would have been nice to have an experiment which directly measures the policy evaluation error. This could be done in a simpler environment where policy evaluation is relatively easy e.g. LQR. This could help support the theory even further.

- How are the curves in figure 1 generated? Are these obtained analytically or are simulations run to obtain then?
It seems like for uniform sampling, if we sample only 1 transition at each step, the expected count (aggregate weight) after $n$ steps would be $\sum_{i=0}^n \frac{1}{i} \approx \log n$. This would imply that the aggregate weight for the transition at time step $i$ would be approximate $\log n - log i$, which seems to roughly match the shape of the curve plotted.

- p.6 The detailed discussion in the "Normalization" paragraph is nice to tie up loose ends.

- p.6-7 Some parts of the "Interpretation" paragraph are a bit confusing to me. In particular, I'm not following the part discussing the bias and variance terms and how they impact the optimization process. I understand that, near the beginning of learning, variance may be large relative to bias. But in phase 2, it's not clear to me what happens to the relative magnitudes. It seems like both bias and variance would be small. Also, while lower policy evaluation error leads to more accurate estimations of $Q^{\pi^N}$, I'm finding it difficult to translate that to how fast optimization progresses. It would seem like more accurate estimates would lead to "better" optimization steps but, at the same time, the optimization process will naturally slow down as the iterates approach a maximum. I think this paragraph should be clarified a bit more.

- Since the policy evaluation error is only one component of the total error $|\hat{Q} - Q^*|$, what do you think about the other term $|Q^{\pi^N} - Q^*|$? Is it not as important to assess the behaviour of the actor-critic algorithms? It seems like focusing on policy evaluation is be enough to understand the performance of actor-critic algorithms in certain cases.

- Does the theory recommend any "optimal weights"? It would be interesting if the derived bounds could be used to develop a better sampling strategy than 1/age or ERE.
- On a similary note, have you tried more aggressive recency weights, such as 1/age^c for c > 1? It would seem to decrease the bias even further although at some cost in the variance. Perhaps an adaptive c would be appropriate too.

- Can the developed theory be used to explain the performance of prioritized ER?


**Summary Of The Paper:**

The paper seeks to improve our understanding of actor-critic methods in deep RL by providing an analysis of the error in estimating the value function. The main theoretical result is an error bound which depends on the Bellman error, variance from sampling and a bias term from the policy mismatch. This error bound suggests that a replay buffer sampling strategy that focuses on recent samples will have an advantage. Experiments validate this hypothesis and using weighted sampling with 1/age weights improves performance. Morover, the theory helps explain the success of the ERE algorithm which also emphasizes recent transitions.


**Summary Of The Review:**

The paper presents a convincing analysis of replay sampling in actor-critic algorithms with theoretical results applicable to modern deep RL algorithms. I think it would be a nice addition to the literature.

---

> ### Author Response · Authors · 2021-11-15
> **A clarification to the questions from Reviewer RYz4**
>
> We thank Reviewer RYz4 for her/his valuable feedback for improving the manuscript. The questions are addressed below.
>
> Q1: p.6 To clarify, why is it stated that $|\hat{Q}-Q^{\pi^N}|$ mainly depends on steps >=i. Is this because of the presence of $\epsilon^{i,L}_{\hat{Q}}$ and $W_1^{i,L}$?
>
> A1: We apologize for the confusion. What we meant was that the difference $|\hat{Q}-Q^{\pi^N}|$, if taking an expectation on the distribution at trajectory step i, mainly depends on steps >=i. This is true because the Q function at step i is determined by the future rewards at steps >=i.
>
> Q2: How are the curves in figure 1 generated?
>
> A2: The curves in Fig 1 are not an analytical form but a numerical one. This is because analytical forms are intractable for most methods (e.g., 1/age and ERE). Uniform sampling is probably the only tractable one as shown in your comment. Thus, we adopt a numerical expression. Below is an example Python code for generating figure 1's uniform & 1/age weighting.
>
>     N = 1000000
>     uni = np.zeros(N)
>     inv_age = np.zeros(N)
>     for n in range(1, N+1):
>         uni[0:n-1] = uni[0:n-1] + np.ones(n)/n
>         hwei = 1./np.arange(n,0,-1)
>         inv_age[0:n-1] = inv_age[0:n-1] + hwei/hwei.sum()
>
> Q3: p.6-7 Some parts of the "Interpretation" paragraph are a bit confusing to me. In particular, I'm not following the part discussing the bias and variance terms and how they impact the optimization process.
>
> A3: We apologize for the confusion. What we meant was that the reduction in policy evaluation errors help the training, and this is visible in phase 1. When the policy evaluation remains low, the training may not improve. Still, we agree that the policy evaluation error is not enough to explain the overall optimization progress, and there are multiple factors when the optimization slows down.
>
> Q4: Since the policy evaluation error is only one component of the total error $|\hat{Q}-Q^*|$, what do you think about the other term $|Q^{\pi^N}-Q^*|$? Is it not as important to assess the behaviour of the actor-critic algorithms? It seems like focusing on policy evaluation is be enough to understand the performance of actor-critic algorithms in certain cases.
>
> A4: We agree that the optimality gap $|Q^{\pi^N}-Q^*|$ is as important as the policy evaluation error. To characterize $Q^*$, we require treatment in the infinity norm. Since the training procedure doesn’t directly optimize the infinity norm, we typically require strong assumptions such as linear MDP to proceed. This is, however, a highly impractical situation. Thus, our discussion currently focuses on the policy evaluation error and we use the reduction on it to explain the optimization progress.
>
> Q5: Does the theory recommend any "optimal weights"? It would be interesting if the derived bounds could be used to develop a better sampling strategy than 1/age or ERE.
>
> A5: The theory currently recommends sampling weights such that the aggregate weights over time are close to uniform, and this is achieved by 1/age or ERE. Although it is difficult to speak about the optimal weights, additional information such as the distance of each historical policy to the current one may help generate better weights.
>
> Q6: Have you tried more aggressive recency weights, such as $1/age^c$ for c > 1? It would seem to decrease the bias even further although at some cost in the variance. Perhaps an adaptive c would be appropriate too.
>
> A6: We focus on 1/age instead of $1/age^c$ because 1/age is more related to ERE and we have more analysis on it. Nevertheless, we did a small experiment on this before. Suppose 0<c<1. There is an improvement of $1/age^c$ for c around 0.8. However, an overly small c makes $1/age^c$ similar to uniform sampling and hence becomes worse than 1/age. Anyway, we believe a slightly varied c around 1; say 0.9 or 0.8, can give a better result.
>
> Q7: Can the developed theory be used to explain the performance of prioritized ER?
>
> A7: The prioritized ER (PER) may help reduce another version of policy evaluation error in the infinity norm. It might have a better connection to the optimal Q function, $Q^*$. Hence, PER is definitely helpful but requires a different interpretation. Since our case mostly works on the 1-norm error, it is not applicable to the explanation of PER.

---

### Decision · Program_Chairs · 2022-01-20

**Decision:**

Reject

**Comment:**

There was some disagreement between reviewers regarding the quality of the paper. Reading the paper, I had difficulty understanding what you were trying to achieve and, similarly to reviewer VgPP, felt the experimental section to be weak. While I can appreciate that compute is expensive, it would have been relevant to design more controllable continuous environments to get cleaner results in addition to those on MuJoCo. As it is, there is a lot of noise (and Table 1 does not contain confidence intervals) which, added to the general brittleness of RL algorithms, makes the experiments lack convincing power.

I encourage the authors to take all the feedback from the authors into account and resubmit an improved version of their work to another conference.